# Deep Multimodal Learning with Missing Modality: A Survey

**Renjie Wu**                                                                *renjie.wu@anu.edu.au*
*The Australian National University*

**Hu Wang**                                                                *hu.wang@mbzuai.ac.ae*
*Mohamed bin Zayed University of Artificial Intelligence*

**Hsiang-Ting Chen**                                                                *tim.chen@adelaide.edu.au*
*Adelaide University*

**Gustavo Carneiro**                                                                *g.carneiro@surrey.ac.uk*
*The University of Surrey*

**Reviewed on OpenReview:** *https://openreview.net/forum?id=tc7RFcx4hT*

## Abstract

During multimodal model training and testing, certain data modalities may be absent due to sensor limitations, cost constraints, privacy concerns, or data loss, which can degrade performance. Multimodal learning techniques that explicitly account for missing modalities aim to improve robustness by enabling models to perform reliably even when certain inputs are unavailable. This survey presents the first comprehensive review of Multimodal Learning with Missing Modality (MLMM), with a focus on deep learning approaches. We outline the motivations and key distinctions between MLMM and conventional multimodal learning, provide a detailed analysis of existing methods, applications, and datasets, and conclude by highlighting open challenges and future research directions.

## 1 Introduction

Multimodal learning has become a crucial field in Artificial Intelligence (AI). It focuses on jointly analyzing various data modalities, including visual, textual, auditory, and sensory information. This approach mirrors the human capacity to combine multiple senses for better understanding and interaction with the environment. Modern multimodal models leverage the robust generalization capabilities of deep learning to uncover complex patterns and relationships that single-modal systems might not detect. This capability is advancing work across multiple domains (Bayoudh et al., 2022; Hu et al., 2023; Streli et al., 2023). Recent surveys show that multimodal learning significantly boosts performance and enables more advanced AI applications (Baltrušaitis et al., 2018; Zhao et al., 2024; Wu et al., 2024a).

However, real-world multimodal systems frequently encounter scenarios where certain data modalities are missing or incomplete. Such cases arise from sensor malfunctions, hardware limitations, privacy concerns, environmental interference, and data transmission failures. Interest in addressing this challenge has grown considerably in recent years, as reflected in the publication trends shown in figure 1a. For instance, in a three-modality setting, data samples can be classified as either full-modality (containing all three modalities) or missing-modality (lacking one or more). Missing modalities can occur at any stage, from data collection to deployment, and can substantially degrade model performance.

Early works in affective computing (Cohen et al., 2004; Sebe et al., 2005) reported that when collecting facial and audio data, some image or audio samples were not useful due to excessive microphone noise or camera obstructions. This has forced them to propose audio-visual models that can handle the missing modality to recognize human emotional states. In the field of medical AI, privacy concerns and the challenges of obtaining certain data modalities during surgeries or invasive treatments often lead to missing modalities in

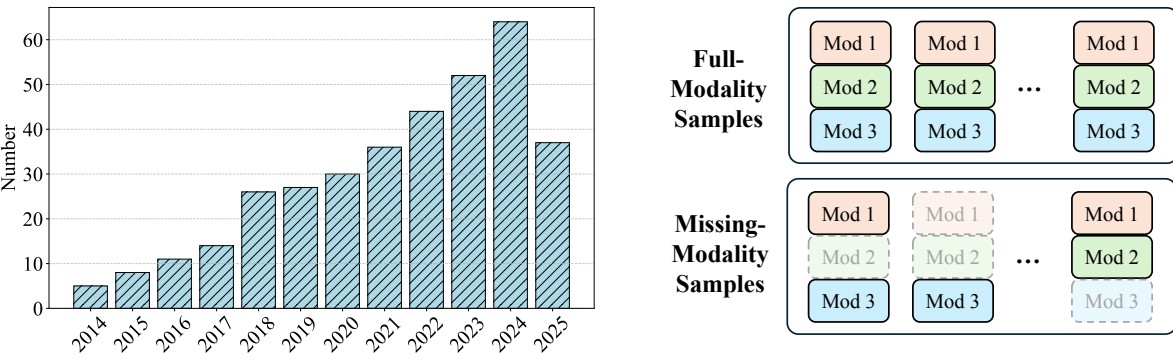

(a) Trends in Missing Modality Publications      (b) Full-Modality vs. Missing-Modality Samples

Figure 1: (a) The trend of papers published on deep multimodal learning with missing modality in the past 10 years (2014-2025.8). The number of publications has increased over time and has received widespread attention from the community. (b) Description of a three-modality scenario with full- and missing-modality samples. We abbreviate "Modality" as "Mod" in all figures of this paper and use dashed boxes with fading colors to represent missing modalities/modules.

multimodal datasets (Zhang et al., 2023b; Li et al., 2025b). Similarly, in space exploration, NASA's Ingenuity Mars helicopter (Donaldson, 2024) faced a missing modality challenge when its inclinometer failed due to extreme temperature cycles on Mars. To address this issue, NASA applied a software patch that modified the initialization of navigation algorithms (Team, 2022). Beyond such domain-specific cases, structural or quantitative heterogeneity across sensor types, versions, or brands can also yield inconsistent modality availability. Combined with the unpredictability of real-world scenarios and the diversity of data sources, these factors make robustness to missing modalities an essential requirement for multimodal systems. Consequently, developing models that can effectively handle incomplete inputs has become a central research focus.

In this survey, we focus on the challenge of handling incomplete inputs in multimodal learning, which we refer to as the **missing modality problem**. We term solutions to this challenge **Multimodal Learning with Missing Modality (MLMM)**. Unlike the conventional full-modality setting, where all **N** modalities are assumed to be consistently available, MLMM must operate when one or more modalities are absent during training or testing. The central goal of MLMM is to robustly exploit the available modalities while maintaining performance comparable to systems trained with complete modality information.

This survey reviews recent advances in MLMM and its applications across diverse domains, including information retrieval (Malitesta et al., 2024; Pipoli et al., 2025), remote sensing (Wei et al., 2023), Pervasive Computing (Nweke et al., 2018; Van Der Donckt et al., 2024; Chen et al., 2023a), robotic vision (Gunasekar et al., 2020; Park et al., 2025), (bio-)medical domain (Zhou et al., 2023; Shah et al., 2023; Azad et al., 2022; Zhu et al., 2025), sentiment analysis (Wagner et al., 2011; Wang et al., 2025b), and multi-view clustering (Chao et al., 2021). We also introduce a fine-grained taxonomy of MLMM methodologies, application scenarios, and corresponding datasets.

**Contributions:** **(1)** A comprehensive survey of MLMM methods across diverse domains, accompanied by an extensive compilation of relevant datasets, highlighting the versatility of MLMM in addressing real-world challenges. **(2)** A novel, fine-grained taxonomy of MLMM methodologies, with a multi-faceted categorization framework based on multi-modal integration stages and missing modality recovery strategies. **(3)** An in-depth analysis of current MLMM approaches, their challenges, and future research directions, contextualized within the proposed taxonomical framework.

**Paper Collection:** In our literature search methodology, we primarily source papers from Google Scholar and major conferences/journals in AI, Machine Learning, Computer Vision, Natural Language Processing, Audio Signal Processing, Data Mining, Multimedia, Medical Imaging, Remote Sensing and Pervasive Com-

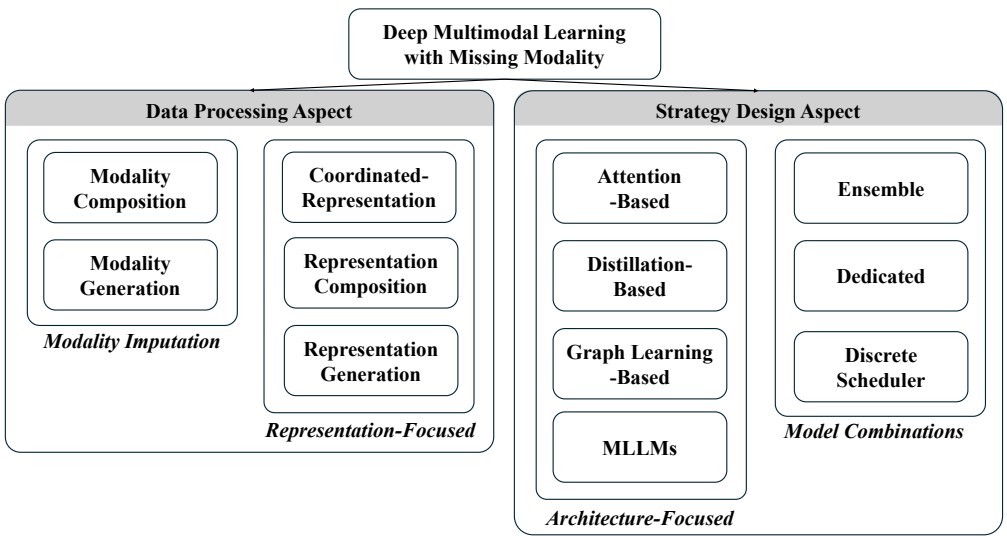

Figure 2: Our taxonomy of deep multimodal learning with missing modality methods. We categorize existing methods into two aspects: data processing and strategy design. **Data Processing**: we differentiate between modality imputation (handling at the modality data level) and representation-focused models (dealing with at the data representation level). **Strategy Design**: we distinguish between architecture-focused models (model architecture adjustments) and model combinations (combining multiple models externally). "MLLMs": multimodal large language models.

puting. The collected papers are from, but not limited to, top-tier conferences (e.g., AAAI, IJCAI, NeurIPS, ICLR, ICML, CVPR, ICCV, ECCV, ACL, EMNLP, KDD, ACM MM, MICCAI, ICASSP) and journals (e.g., TPAMI, TIP, TMI, TMM, JMLR, TMLR). Please refer to Section A in the Supplementary Materials for the full names of these conferences and journals. We have compiled a total of 354 significant papers from the period spanning 2012 to August 2025. Our search strategy involved using keywords such as "incomplete," "missing," "partial," "absent," and "imperfect," combined with terms like "multimodal learning," "deep learning," "representation learning," "multi-view learning," and "neural networks."

**Survey Organization:** Firstly, we explain the background and motivation of this survey in section 1. In section 2, we introduce our taxonomy and categorize existing deep MLMM methods from a methodological perspective, detailing them in two aspects and four types (figure 2). In the following section 3 and section 4, we introduce various methods from the aspects of model data processing and strategy design respectively. We then summarize current application scenarios and corresponding datasets used in this field in section 6. In section 7, we discuss unresolved challenges and future directions. Finally, we present our conclusions drawn from the exploration of deep MLMM in section 8.

## 2 Methodology Taxonomy: An Overview

We review current deep MLMM methods from two key aspects: data processing and strategy design, based on the contributions of existing work.

### 2.1 Data Processing Aspect

Methods that focus on exploring the data processing aspect of a model or framework can be divided into *Modality Imputation* and *Representation-Focused Models*, depending on whether the model's handling of missing modalities occurs at the modality data level or at the data representation level.

**(1) Modality Imputation** operates at the modality data level and fills in the missing information by compositing (Chen et al., 2020; Sun et al., 2024b; Parthasarathy & Sundaram, 2020; Yang et al., 2024;

Campos et al., 2015) (modality composition methods) or generating (Chen et al., 2024a; Ma et al., 2021c; Arya & Saha, 2021; Wang et al., 2024c; Tran et al., 2017; Zhang et al., 2023c) absent modalities (modality generation methods) from available modalities. Those approaches are rooted in the idea that if a missing modality can be accurately imputed, downstream tasks can continue as if "full" modalities were available.

**(2) Representation-Focused Models** are designed to address missing modalities at the representation level. In some cases, the coordinated representation methods (Liang et al., 2019; Wang et al., 2020b; Ma et al., 2021b) impose some specific constraints to help align the representations of available modalities in the same semantic space, so that the model can be effectively trained even when facing missing modalities. Other representation-focused methods either generate the missing-modality representation using available data (Zheng et al., 2021; Li et al., 2024b; Hoffman et al., 2016; Ma et al., 2021c; Park et al., 2024) or combine the representations of existing modalities (Zhi et al., 2024; Sun et al., 2024b; Zhou et al., 2021; Delgado-Escano et al., 2021; Zheng et al., 2024; Zhang et al., 2024) to fill in the gaps.

## 2.2 Strategy Design Aspect

Methods that explore the strategy design aspect are based on models that can dynamically adapt to different missing-modality cases during training and testing through flexible adjustments of the model architecture (internal model architecture adjustment) and the combination of multiple models (external model combinations). We name them *Architecture-Focused Models* and *Model Combinations*.

**(1) Architecture-Focused Models** address missing modalities by designing flexible model architectures that can adapt to varying numbers of available modalities during training or inference. A key technique here is based on attention mechanisms (Ge et al., 2023; Yao et al., 2024; Gong et al., 2023; Mordacq et al., 2024; Radosavovic et al., 2024; Jang et al., 2024; Qiu et al., 2023), which dynamically adjusts the modality fusion and processing, allowing the model to handle any number of input modalities. Another approach is based on knowledge distillation (Wang et al., 2020a; Saha et al., 2024; Zhang et al., 2023b; Chen et al., 2024b; Wang et al., 2023a; Shi et al., 2024; Wang et al., 2023c), where the model is trained to accommodate missing modalities by transferring knowledge from full-modality models to those operating with incomplete data or distilling between different branches inside the model. Additionally, graph learning-based methods (Zhao et al., 2022; Zhang et al., 2022; Lian et al., 2023; Malitesta et al., 2024) exploit the natural relationships between modalities, using graphs to dynamically fuse and process available modalities while compensating for missing ones. Finally, MLLMs (Zhan et al., 2024; Wu et al., 2023b; Li et al., 2023a) also play a crucial role in this category, as their ability to handle long contexts and act as feature processors enables them to accept and process representations from any number of modalities. These architectural strategies collectively allow models to maintain performance even when dealing with incomplete multimodal inputs.

**(2) Model Combinations** tackle missing modality problems by employing strategies that leverage multiple models or specialized training techniques. One approach is to use dedicated training strategies (Zeng et al., 2022; Chen et al., 2022; Xue & Marculescu, 2023) tailored for different modality cases, ensuring that each case is trained for optimal performance. Another approach involves ensemble methods (Hu et al., 2020; Wang et al., 2021; Lee et al., 2023b), where models trained on either partial/full sets of modalities are combined, allowing the system to select the most suitable model based on the available modalities to do joint predictions. Additionally, discrete scheduler methods (Wu et al., 2023a; Shen et al., 2024; Surís et al., 2023) can incorporate various downstream modules to flexibly process any number of modalities and handle specific tasks. These schedulers intelligently select and combine the outputs of multiple models or modules to manage missing-modality scenarios, offering a versatile solution for multimodal tasks.

Our taxonomy (figure 2) can reflect different aspects and levels of multimodal learning—ranging from modality data to data representations, architectural design, and model combinations—each providing a distinct way to approach the problem of missing modalities based on the task requirements and available resources.

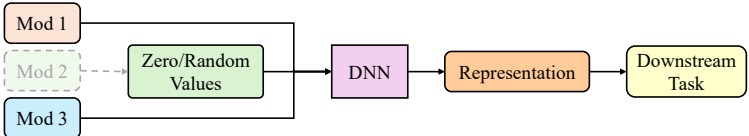

Figure 3: Zero/Random values composition methods. If we assume modality 2 is missing, then this modality will be replaced with zero/random values. "DNN" in all figures of this survey means different kinds of deep neural networks.

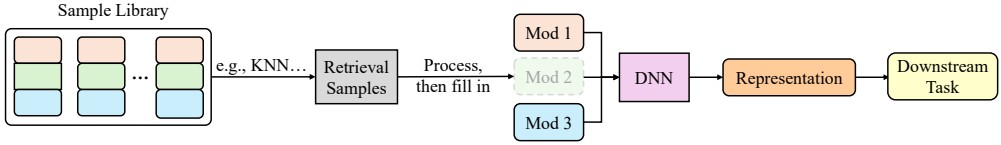

Figure 4: Retrieval-based modality composition methods search for one or more samples by randomly selecting or using simple retrieval algorithms like KNN, or its variants, from same-category samples that have the required missing modalities, and then compose them with the input missing-modality sample to form a "full"-modality sample.

## 3 Methodologies in Data Processing Aspect

### 3.1 Modality Imputation

Modality imputation refers to a technique used by MLMM methods that fills in missing-modality samples or generates missing modalities to complete the dataset with missing modalities by performing various transformations or operations on existing modalities. Modality imputation methods addressing the missing modality problem at the data modality level can be categorized into two types. (1) Modality composition methods use zero/random values or data copied from similar instances as the input for the missing modality data. The data produced via these methods to represent the missing data are then composited with the data from the available modalities to form a "full"-modality sample. (2) Modality generation methods generate the missing modality data using generative models such as Auto-Encoders (Hinton & Zemel, 1993), Generative Adversarial Networks (GANs) (Goodfellow et al., 2014), or Diffusion Models (Ho et al., 2020). The generated data is then composited with the data from the available modalities to form a "full"-modality sample. We provide more details about these two methods in the next sub-sections.

#### 3.1.1 Modality Composition Methods

Modality composition methods are widely employed for its simplicity and effectiveness to maintain the original dataset size. Zero/Random Values Composition Methods represent a type of modality composition method that replaces a missing modality with zeros or random values, as shown in figure 3. In recent research (Chen et al., 2020; Sun et al., 2024b; Liu et al., 2023a; Malitesta et al., 2024), they are often used as a baseline method for comparison with other more sophisticated methods. For the missing sequential data problem, such as missing frames in videos, the similar Frame-Zero method (Parthasarathy & Sundaram, 2020) was proposed to replace the missing frames. These methods are prevalent in typical multimodal learning procedures and can be used to balance and integrate information from different modalities when making predictions. This prevents the model from over-relying on dominant modalities available for each sample, enhancing its robustness by encouraging a more balanced integration of information across all available modalities.

Retrieval-Based Representation Composition Methods (figure 4) represent another type of modality composition method, which replaces the missing modality data by copying or averaging data from retrieved samples with the same classification. Some other methods randomly select a sample that has the same classification and required missing modality from other samples. The selected modality data is then composed with the missing-modality sample to form a full-modality sample for training. But those retrieval-based modality

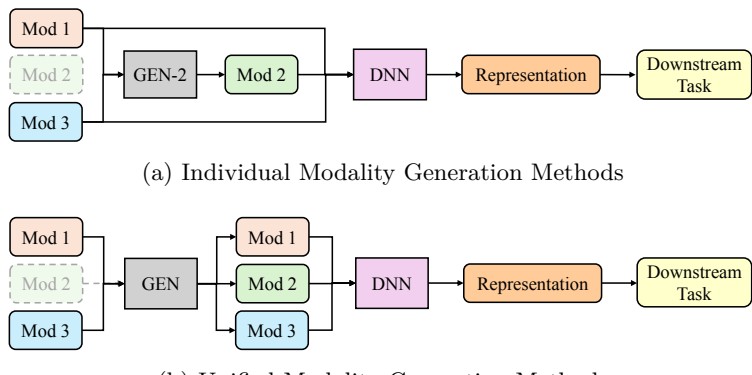

(a) Individual Modality Generation Methods

(b) Unified Modality Generation Methods

Figure 5: Description of two typical modality generation methods. We set modality 2 as the missing modality for examples and use other available modalities to generate modality 2. "GEN" in both figures represents modality generation networks. (a) We set up a modality-2 generator (GEN-2) from other modalities. (b) All modalities are input and generated together by using a single GEN.

composition methods are not applicable to pixel-level tasks, such as segmentation, and are only suitable for simple tasks (e.g., classification) because they may lead to the overfitting of noisy data, if mismatched samples are combined. For example, Yang et al. (2024) proposed Modal-mixup to complete the training datasets by randomly complementing same-category samples with missing modalities. However, such methods cannot solve the missing modality problem during the testing phase because they rely on known labels of training data samples. In some multimodal streaming data classification tasks, such as audio-visual expression recognition, video streams may experience frame drops due to network communication packet loss, etc. Frame-Repeat (Parthasarathy & Sundaram, 2020) was proposed to to make up for the missing frames by using past frames.

Other methods (Yang et al., 2024; Campos et al., 2015) also used K-Nearest Neighbors (KNN), or its variants, to retrieve the best-matched samples for composition. For these matched samples, they select the sample with the highest score or obtain the average values of these samples to supplement the missing modality data. Their experiments have shown that KNN-based methods generally perform better than the above methods, and can handle missing modality during testing. Nevertheless, such KNN retrieval-based modality composition methods often suffer from high computational complexity, sensitivity to imbalanced data, and significant memory overhead. In pervasive computing, some researcher (Hussein et al., 2024a) proposed to cluster sensor data and pre-learns the best imputation pattern for each cluster combination, enabling fast lookup-based completion of missing modality at runtime.

All methods above can complete datasets with missing modalities, but they reduce the diversity of the dataset because they may introduce duplicated training samples. This is especially problematic with datasets having a high rate of modality missing, where most samples have missing-modality data, as it increases the risk of overfitting to certain classes with few full-modality samples, if they pad missing-modality samples with duplicated ones.

### 3.1.2 Modality Generation Methods

With deep learning, synthesizing missing modalities has become more effective by leveraging powerful representation learning and generative models that can capture complex cross-modal relationships. Current methods for generating missing-modality data are divided into individual and unified generative methods.

Individual Modality Generation Methods train an individual generative model for each modality in case any modality is missing, as shown in figure 5a. Early works used Gaussian processes (Mario Christoudias et al., 2010) or Boltzmann machines (Sohn et al., 2014) to generate missing modalities from available data. With deep learning, models like Auto-Encoders (AEs) (Hinton & Zemel, 1993) can be used for missing modality generation. Li et al. (2014) used 3D-CNN to generate positron emission tomography (PET) data

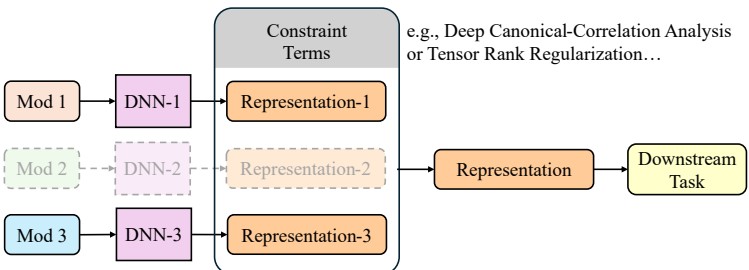

Figure 6: Illustration of coordinated-representation methods, which impose constraints to make the learned representations semantically consistent. Assuming that modality 2 is missing, these methods use constraints terms between the features of different modalities.

from magnetic resonance imaging (MRI) inputs. Chen et al. (2024a) addressed missing modalities in MRI segmentation by training U-Net models to generate other two modalities from MRI. A recent work (Ma et al., 2021c) used AEs as one of the baselines to complete datasets by training one AE per modality. In domain adaptation, Zhang et al. (2021) proposed a U-Net-based Multi-Modality Data Generation module with domain adversarial learning to generate each missing modality by learning domain-invariant features. In immunohistochemical, DeepLIIF (Ghahremani et al., 2022) uses ResNet to generate the various modalities such as Hematoxylin, mpIF DAPI, mpIF Lap2, and mpIF Ki67. HE2RNA (Schmauch et al., 2020) is a cross-modal model that predicts gene expression (transcriptomic modality) from H&E whole-slide images (visual modality), generates spatialized gene-expression heatmaps, and transfers this representation to downstream clinical tasks.

GANs significantly improved image generation quality by using a generator to create realistic data and a discriminator to distinguish it from real data. Researchers therefore began replacing above models with GANs for modality generation. For example, GANs generate missing modalities using latent representations of existing ones in breast cancer prediction (Arya & Saha, 2021). In remote sensing, Bischke et al. used GANs to generate depth data, improving segmentation over RGB-only models (Bischke et al., 2018). GANs were also used in robotic object recognition models to help generate missing RGB and depth images (Gunasekar et al., 2020). Recent studies (Ma et al., 2021c) show that GANs outperform AEs in generating more realistic missing modalities and can lead to better downstream-task model performance. SensorGAN (Hussein & Bhat, 2024) introduces a GAN model that restores absent sensor channels, directly targeting missing-modality recovery in wearable human-activity-recognition pipelines. Recently, the introduction of Diffusion models has further improved image generation quality. Wang et al. proposed the IMDer method (Wang et al., 2024c), which uses available modalities as conditions to make diffusion models generate missing modalities. Experiments showed it can reduce semantic ambiguity between recovered and missing modalities and achieves good generalization performance than previous works.

Unified Original Data Generative Methods train a unified model that can generate all modalities simultaneously (figure 5b). One representative model is Cascade AE (Tran et al., 2017), which stacks AEs to capture the differences between missing and existing modalities for generating all missing modalities. Recently, Zhang et al. (2023c), have attempted to use attention and max-pooling to integrate features of existing modalities, enabling modality-specific decoders to accept available modalities and generate other missing modalities. Experiments demonstrated that this method is more effective than using max-pooling alone (Chartsias et al., 2017) to integrate features from any number of modalities for generating all missing modalities together. Above methods for generating missing modalities can mitigate performance degradation to some extent. However, when facing a scenraio where one of the modalities is severely missing, training a generator that can produce high-quality missing modalities remains challenging. Additionally, the modality generation model significantly increases storage and computational requirements. As the number of modalities grows, the complexity of these generative models also increases, further complicating the training process and resource demands.

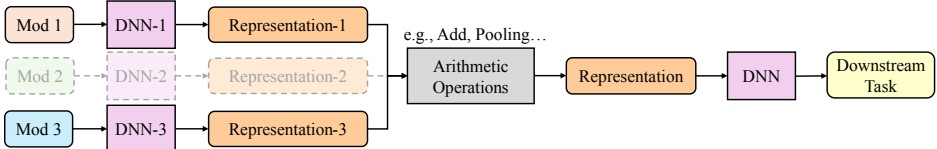

Figure 7: The general idea of the arithmetic operation-based representation composition methods. The representation of any number of modalities can be combined to a fixed dimension that is the same as the dimension required by the subsequent DNN layer.

## 3.2 Representation-Focused Models

Representation-focused models address the missing modality problem at the representation level. We introduce such models by first presenting two coordinated-representation-based approaches that enhance the learning of more discriminative and robust representations by imposing specific constraints (figure 6). The next type of representation-focused methods that we discuss are the representation imputation methods, which can be categorized into representation composition and representation generation methods. Representation composition methods can borrow the solutions described in section 3.1.1 and operate at the representation level of the modalities or employs arithmetic operations (e.g., pooling) to fuse a dynamic number of modalities. Finally, we introduce the representation generation methods, which usually use small generative models to produce the representations of missing modalities.

### 3.2.1 Coordinated-Representation Methods

Coordinated-representation methods focus on introducing certain constraints between representations of different modalities to make the learned representations semantically consistent. They are usually divided into two categories, one based on regularization and the other driven by correlation. An example of regularization is Tensor Rank Regularization (TRR) (Liang et al., 2019). TRR achieves multimodal fusion by taking the outer products of different-modality tensors and taking the sum of all their outer products. In order to solve the high rank of multimodal representation tensors caused by imperfect modalities (missing or noisy) in time series, Paul et al. introduced tensor rank minimization to try to keep the multimodal representation low rank to better express the tensors of true correlation and potential structure in multimodal data, thereby alleviating the imperfection of the input.

Some methods train models by coordinating the correlation between different modal features. For example, Wang et al. (2020b) use a Deep Canonical Correlation Analysis (CCA) module to maximize feature associations of available modalities by using Canonical Correlation Coefficient, which enables training on incomplete datasets. Ma et al. (2021b) proposed a Maximum likelihood function to characterize conditional distributions of full-modality and missing-modality samples during training. Other research efforts (Liu et al., 2021b) have added constraints based on the Hilbert-Schmidt Independence Criterion (HSIC), which helps models learn how to complete missing modality features by enforcing independence between irrelevant features while maximizing the dependence between relevant ones. These methods (Matsuura et al., 2018; Ma et al., 2021a; Li et al., 2022a) aid models in learning how to complete missing modality features or training on incomplete datasets by learning the similarity or correlation between features. A drawback of the above methods is that they perform well only when two or three modalities are used as input (Zhao et al., 2024), and their effectiveness tends to degrade as the number of modalities increases. Even when trained on a dataset with missing modalities, some methods still struggle to effectively address unseen missing modality combinations or cases during testing.

### 3.2.2 Representation Composition Methods

There are two types of representation composition methods, which are explained below. Retrieval-Based Representation Composition Methods attempt to recover the missing modality representation by retrieving the modality data from existing samples, similarly to the modality composition method in section 3.1.1. They typically use pre-trained feature extractors to generate features from available

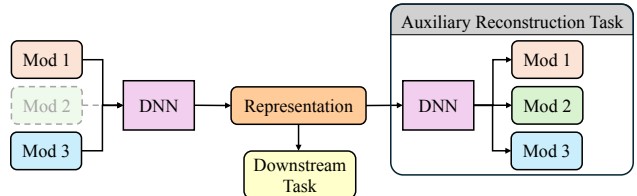

(a) Indirect-to-Task Representation Generation Methods

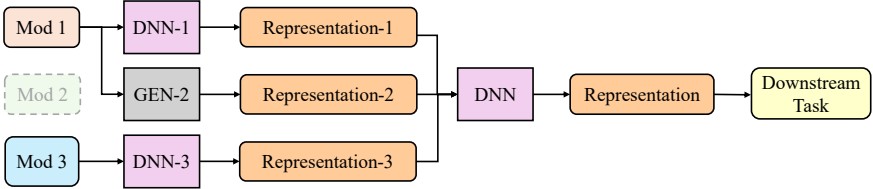

(b) Direct-to-Task Representation Generation Methods

Figure 8: Description of two typical representation generation methods. We assume modality 2 is missing here. (a) Indirect-to-task representation generation methods are supervised by the loss of two tasks, namely: auxiliary reconstruction and downstream tasks. Note that these two tasks sometimes can be trained separately, first training the reconstruction task, discarding the generator, and then training the downstream task, such as the training progress of ActionMAE (Woo et al., 2023). While (b) direct-to-task representation generation methods directly use a modality generation network ("GEN-2" in figure) to generate representations of the missing modality 2 during training and inference without reconstructing the modality 2.

samples, storing them in a feature pool (Zhi et al., 2024; Sun et al., 2024b). Cosine similarity is then used to retrieve matched features for the input sample, with the average of these missing-modality features in top-$K$ samples filling the representation of missing modalities. Additionally, some methods, such as Missing Modality Feature Generation (Wang et al., 2023b), replace missing modality features by averaging the representations of available modalities, assuming similar feature distributions across modalities.

Arithmetic Operation-Based Representation Composition Methods can flexibly fuse any number of modality representations through arithmetic operations such as pooling without learnable parameters (figure 7). Researchers (Zhou et al., 2021; Wang et al., 2023b) have fused features through operations like average/max pooling or addition, offering low computational complexity and efficiency. However, drawbacks include potential loss of important information. To address this, merge operations (Delgado-Escano et al., 2021) use a sign-max function, which selects the highest absolute value from feature vectors, yielding better results by preserving both positive and negative activation values. TMFormer (Zhang et al., 2024) introduces token merging based on cosine similarity. Similar approaches (Zheng et al., 2024) focus on selecting key vectors or merging tokens to handle missing modalities. A recent work (Sarkar et al., 2025) propose to perform weighted fusion of these pooled features to learn a fixed-size multimodal embedding

Therefore, those representation composition approaches not only allow the model to flexibly handle features from any number of modalities but also enable learning without introducing new learnable parameters. However, they are not good at capturing the inter-modality relationships through learning, as methods like selecting the largest vector or the feature with the highest score is difficult to fully represent the characteristics of all modalities.

### 3.2.3 Representation Generation Methods

Compared to modality generation methods in section 3.1.2, representation generation methods can be integrated seamlessly into existing multimodal frameworks. Current methods fall into two categories: (1) Indirect-to-task representation generation methods (figure 8a) treat modality reconstruction as an auxiliary task during training, helping the model intrinsically generate missing modality representations for downstream tasks. Since the auxiliary task aids in representation generation during training but is dropped

during inference of downstream task, it is termed "indirect-to-task." (2) Direct-to-task representation generation methods (figure 8b) train a small generative model to directly map available modality representations to the missing modalities. We provide more details about these two categories of methods below.

Indirect-to-Task Representation Generation Methods can indirectly generate missing modality representations and often employ the "encoder-decoder" architecture. During training, modality-specific encoders extract features from available modalities, reconstruction decoders reconstruct the missing modality, and the downstream module is supervised by downstream task loss for prediction. Those reconstruction decoders are discarded during inference, where predictions rely on both the generated missing modality representations and the existing ones. This approach typically employs Multi-Task Learning, training both downstream tasks and auxiliary tasks (modality reconstruction) simultaneously. Some methods, inspired by Masked Autoencoder (He et al., 2022), split reconstruction and downstream task training. Typically, in the training process of this type of methods, the features of the available modalities are usually input into the downstream task module and the reconstruction decoder of different modalities after being fused. However, some methods accept the outputs of different modality encoders into their respective reconstruction decoders. Based on this distinction, we divide this type of methods into before-fusion and after-fusion methods.

*Before Fusion* methods receive the features from modality-specific encoders directly into each specific reconstruction decoder for missing modality reconstruction. For example, MGP-VAE (Hamghalam et al., 2021) uses output of the VAE encoder of each modality as the input of each reconstruction decoder and the fused output for Gaussian Process prediction. Similarly, PFNet (Zheng et al., 2021) employs RGB and thermal-infrared modalities for pedestrian re-identification, feeding outputs of each encoder into corresponding decoders before fusing them together. Li et al. (2024b) adopted a similar approach for brain tumor segmentation, reconstructing each modality before fusing encoder outputs.

*After Fusion* methods use the fused features from modality-specific encoders as the input of reconstruction decoder. Tsai et al. (2018) introduced a Multimodal Factorization Model where modality-specific encoder-decoders reconstruct the missing modality, while the downstream task module employs a separate multimodal encoder-decoder. To recover the missing modality features, the outputs of the downstream task model's encoder are not only used for predictions of the downstream task decoder, but also fused with the outputs of each modality-specific encoder for inputting to the each corresponding reconstruction decoder. Chen et al. (2019) and Li et al. (2025a) also decoupled the encoders for reconstruction and downstream tasks, showing that disentangling features helped eliminate irrelevant features. Jeong et al. (2022) proposed a simpler approach using multi-level skip connections in a U-Net architecture to fuse features for missing modality reconstruction, achieving better generalization in brain tumor segmentation than (Chen et al., 2019). Further, Zhou (2023) and Sun et al. (2024a) demonstrated effective cross-modal attention fusion techniques could help enhance reconstruction and segmentation. Most of these methods train reconstruction and downstream tasks concurrently. However, some approaches, such as ActionMAE (Woo et al., 2023) and M3AE (Liu et al., 2023a), first train reconstruction tasks and then fine-tune the pre-trained encoders for downstream tasks.

Although these methods achieve strong performance, they still require the missing modality data to be present during training to supervise the reconstruction decoders.

Direct-to-Task Representation Generation Methods usually use simple generative models to directly map information from avaiable modalities to the missing modality representation. The general concept is illustrated in figure 8b. Early work (Mario Christoudias et al., 2010) using Gaussian Processes to "hallucinate" missing modalities inspired Hoffman et al. (Hoffman et al., 2016) to propose a Hallucination Network (HN) that predicts missing depth modality features from RGB images. The HN aligns intermediate layer features with another network (trained on depth images) using an L2 loss, allowing RGB inputs to generate depth features to address missing modality challenges. This method has been extended to pose estimation (Choi et al., 2017) and land cover classification (Kampffmeyer et al., 2018), where the HN can be used to generate missing heat distribution and depth features.

In tasks like sentiment analysis, a translation module can map available-modality representations to the missing ones, with L2 loss applied to align outputs with those from networks trained on missing modalities (Huan et al., 2023). A notable recent work, SMIL (Ma et al., 2021c), employs Bayesian learning and

meta-regularization to directly generate missing modality representations. SMIL has shown strong generalization on datasets with a high rate of missing modality samples. Subsequent work (Li et al., 2022d) addressed challenges such as distribution inconsistency, failure to capture specific modality information, and lack of correspondence between generated and existing modalities. Researchers have also explored cross-modal distribution transformation methods to align distributions of generated and existing modality representations, enhancing discriminative ability (Wang et al., 2023d). In audio-visual question answering (AVQA), Park et al. (2024) proposed a Relation-aware Missing Modal Generator (RMM) that consist of to generate pseudo features of missing modalities, improving robustness and accuracy. The RMM consists of a visual generator, auditory generator, and text generator, which generate missing auditory representations by analyzing the correlations between visual and textual modalities and utilizing learnable parameter vectors (slots) for reconstruction and synthesis of missing-modality features.

Recently, to address the potential absence of different modalities in sentiment analysis, Guo et al. (2024) proposed a Missing Modality Generation Module for prompt learning. This module maps prompts of available modalities to prompts of missing modalities through a set of projection layers. Lin et al. (2024) proposed Uncertainty Estimation Module which can identify useless tokens in different modality tokens to facilitate better use of U-Adapter-assisted pre-trained models to better utilize the tokens of available modalities when downstream tasks face missing modalities. Liao et al. (2025) explore a variety of different conditions of sensor impairment are and the capabilities of various existing multimodal segmentation models are evaluated. Park et al. (2025) use the Multi-modal Mixture of Expert, which not only enables the model to dynamically handle multi-modality, but also allows different branches to focus on their own modalities by sharing a query.

Compared to indirect methods, direct generation methods avoid the cumbersome reconstruction of modality data, using feature generators that are easier to train and integrate into multimodal models.

In summary, representation generation methods are limited by the imbalance of missing modalities in the training dataset or the limited number of full-modality samples, which may lead to training procedures that overfit to the existing modalities. In addition, indirect-to-task representation generation methods can access complete multimodal data samples during training, which means they can simulate arbitrary missing modality cases during training by dropping modalities. As a result, they generally achieve better performance than direct-to-task methods, which must handle incomplete multimodal data samples during training.

## 4 Methodologies in Strategy Design Aspect

### 4.1 Architecture-Focused Models

Different from above methods of handling missing modality problems at the modality or representation level, many researchers adjust the model training or testing architecture to adapt to the missing-modality cases. We divide them into four categories according to their core contributions in dealing with missing modalities: attention-based methods, distillation-based methods, graph learning-based methods, and multimodal large language models.

#### 4.1.1 Attention Based Methods

In the self-attention mechanism (Vaswani et al., 2017), each input is linearly transformed to generate `Query`, `Key`, and `Value` vectors. Attention weights are computed by multiplying the query of each element with the keys of others, followed by scaling and softmax to ensure the weights sum to 1. Finally, a weighted sum of the values generates the output. We classify attention-based MLMM methods into two categories. (1) Attention fusion methods put attention on modality fusion, integrating multimodal information, which do not rely on any specific model type and can be suitable for various model types as its input and output dimensions are the same. (2) Others are transformer-based methods, which stack attention layers to handle large-scale data with global information capture and parallelization. We provide more details of these two methods below.

Attention Fusion Methods have the powerful ability to capture key features and can be seen as plug-and-play modules. We categorize them into two types: intra- and inter-modality attention methods. Intra-

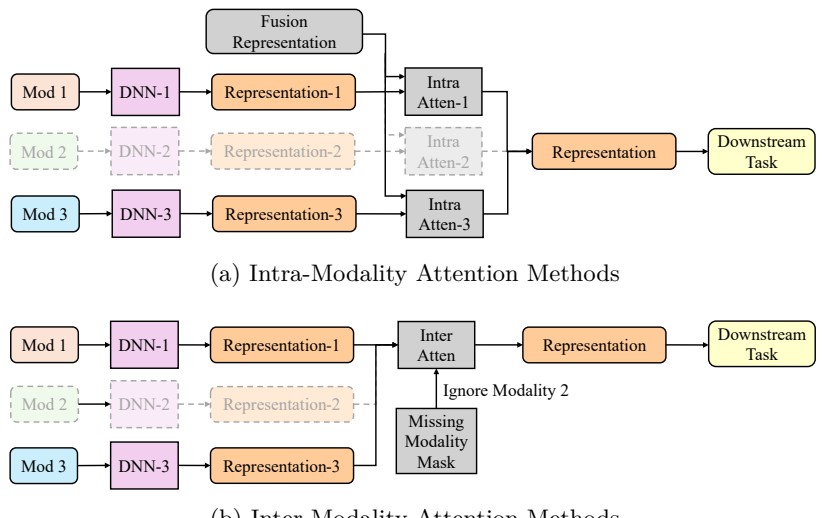

Figure 9: Two typical attention fusion methods, assuming modality 2 as the missing modality. (a) The intra-modality attention ("Intra-Atten" in figure) of missing modality 2 can be skipped. Since each Intra-Atten of different modalities shares the fusion representation (like learnable query in (Ge et al., 2023) or bottleneck tokens in (Nagrani et al., 2021)), after calculating them one by one, the output representation can be regarded as the fusion of available modalities. (b) "Missing Modality Mask" is the custom mask that is generated according to the missing modality 2 and can help inter-modality attention ("Inter-Atten" in figure) ignore the missing modality 2. By setting the tokens of modality 2 to zero or negative infinity through masking, we can force the attention mechanism to ignore the missing modality 2.

Modality Attention Methods compute attention for each modality independently before fusing them, as shown in figure 9a. This approach focuses on relationships within a single modality, and the fusion between modalities is achieved by sharing partial information. For instance, in 3D detection, the BEV-Evolving Decoder (Ge et al., 2023) handles sensor failures by sharing same BEV-query with each modality-specific attention modules, allowing fusion of any number of modalities. Similarly, in clinical diagnosis, Lee et al. (2023a) proposed modality-aware attention to perform intra-modality attention and predict decisions by using shared bottleneck fusion tokens (Nagrani et al., 2021). Inter-Modality Attention Methods, often based on masked attention, treat missing modality features as masked vectors (using zero or negative infinity values) to better capture dependencies across available modalities, as illustrated in figure 9b. Unlike conventional cross-modal attention, masked attention models share the same parameters across all embeddings, allowing flexible handling of missing modalities. For example, Qian & Wang (2023) designed an attention mask matrix to ignore missing modalities, improving model robustness. Similarly, DrFuse (Yao et al., 2024) decouples modalities into specific and shared representations, using the shared ones to replace missing modalities, with a custom mask matrix to help model ignore the specific representations of the missing modality.

Transformer Based Methods can be divided into two types: joint representation learning (JRL) and parameter efficient learning (PEL) according to full parameter training and a small amount of parameter fine-tuning. Since transformers have long context lengths to handle many feature tokens, the multimodal transformer can learn joint representations from any number of modality tokens (figure 10). Some methods first use modality encoders to extract feature tokens from available modalities and then feed them into a multimodal transformer. Gong et al. (2023) introduced an egocentric multimodal task, proposing a transformer-based fusion module with a flexible number of modality tokens and a cross-modal contrast alignment loss to map features into a common space. Similarly, Mordacq et al. (2024) leveraged Masked Multimodal Transformers, treating missing modalities as masked tokens for robust JRL. Ma et al. (2022) proposed an optimal fusion strategy search strategy within the multimodal transformer to help find the best fusion method for different missing/full-modality representations. Radosavovic et al. (2024) introduced specially designed mask

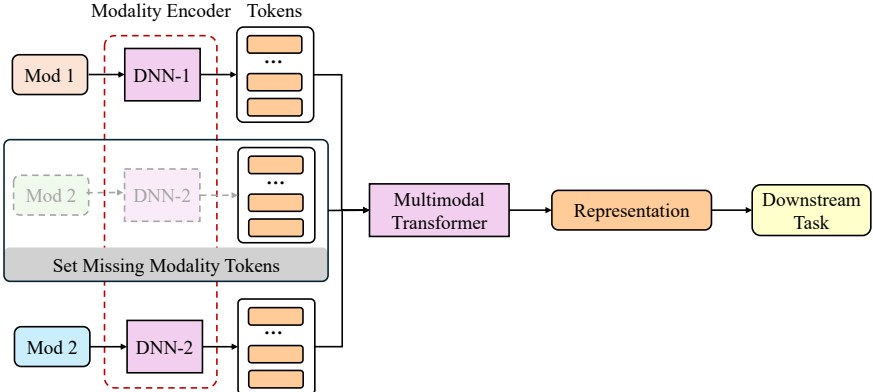

Figure 10: Description of general idea of joint representation learning methods, assuming Modality 2 is the missing modality. The missing modality 2 tokens can be replaced by specific masked tokens. The modality encoders (red dashed box) can be replaced by linear projection layers in some methods.

tokens for an autoregressive transformer to manage missing modalities, successfully deploying the model in real-world scenarios.

With the rise of pre-trained transformer models, PEL methods have been developed to fine-tune these models by training few parameters. Two common PEL methods for pre-trained models are prompt and adapter tuning. Initially used in natural language processing, prompt tuning optimizes input prompts while keeping model parameters fixed. It has since been extended to multimodal models. Jang et al. (2024) introduced modality-specific prompts to address limitations in earlier methods, which merge when all modalities are present and allow the model to update all learnable prompts during training. Liu et al. (2024b) further improved this by proposing Fourier Prompt, which uses fast Fourier transform to encode global spectral information of available modalities into learnable prompt tokens which can be used to supplement missing-modality features, enabling cross-attention with feature tokens to address missing modalities. Adapter tuning, on the other hand, involves inserting lightweight adapter layers (e.g., MLPs) into pre-trained models to adapt to new tasks without modifying the original parameters. Qiu et al. (2023) proposed a method that used a classifier to identify different missing-modality cases and used intermediate features of that classifier as the missing-modality prompt to cooperate with lightweight Adapters to address missing modality problems.

Although attention-based fusion mechanisms in above methods can effectively help deal with the missing modality problem in any framework, none of them cares about the missing modality that may contain important information required for prediction. In the transformer-based methods, the JRL methods are often limited by a large amount of computing resources and require relatively large datasets to achieve good performance. The PEL methods can achieve efficient fine-tuning, but their performance is still not comparable to that of JRL.

### 4.1.2 Distillation Based Methods

Knowledge distillation (Hinton et al., 2015) transfers knowledge from a teacher model to a student model. The teacher model, with access to more information, helps the student reconstruct missing modalities. Below, we categorize two types of distillation methods for addressing this problem.

Representation-Based Distillation Methods transfer rich representations from the teacher model to help the student capture and reconstruct missing modality features. We classify them based on whether they use logits or intermediate features. figure 11 illustrates this type of method. Response distillation methods focus on transferring teacher models' logits to students, helping it mimic probability distributions. Wang et al. (2020a) trained modality-specific teachers for missing modalities, then used their soft labels to guide a multimodal student. Hafner & Ban (2023) employed logits from a teacher model trained with optical data to supervise a reconstruction network for approximating missing optical features from radar data. A

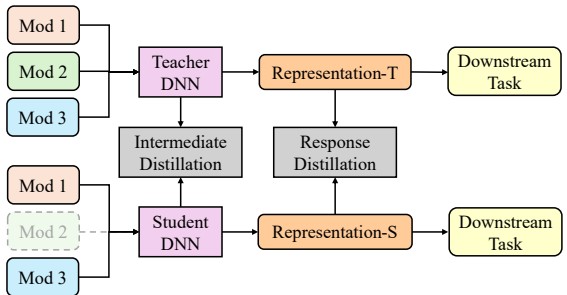

Figure 11: Description of representation distillation methods. Here we set modality 2 is missing. In order to distinguish the representations of teachers and students, we add "-T" and "-S" after their representations in the figure. We refer to intermediate distillation as using features from any combination of intermediate layers within models.

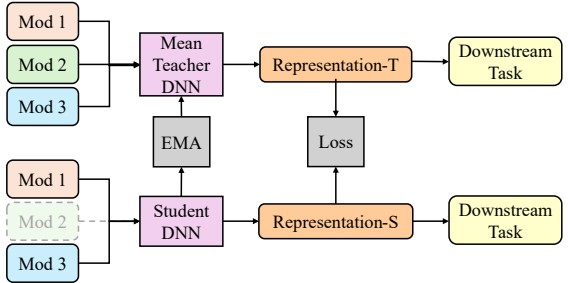

(a) Mean Teacher Distillation Methods

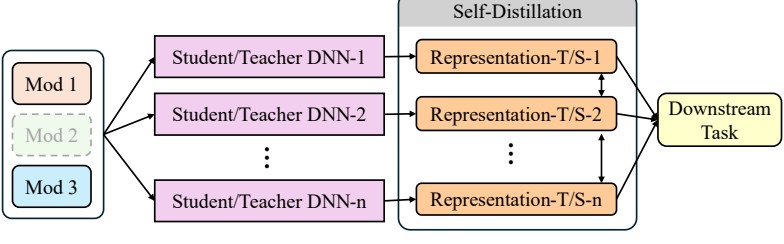

(b) Self Distillation Methods

Figure 12: Description of two process-based distillation methods. (a) "EMA" means exponential moving average. To demonstrate the general architecture of Mean Teacher Distillation, we set the loss on the logits representation. (b) Self-distillation methods for missing modality problems involve the use of models that act as teachers and students, where these models use their own soft labels/representations from each branch to refine themselves during training.

Modality-Aware Distillation was proposed by Saha et al. (2024), leveraging both global and local teacher knowledge in a federated learning setting to help student models to learn how to handle missing modalities.

Intermediate distillation methods align intermediate features between teacher and student models. Shen & Gao (2019) used Domain Adversarial Similarity Loss to align intermediate layers of the teacher and student, improving segmentation in missing modality settings. Zhang et al. (2023b) applied intermediate distillation in endometriosis diagnosis by distilling features from a TVUS-trained teacher to a student using MRI data. Some researchers propose MultiPro (Xu et al., 2025) to learn the distribution of the missing modality by distilling unimodal knowledge into learnable unimodal prompts for each modality from all remaining samples available of this modality. Some also tackles the missing-modality problem in multimodal federated mobile sensing systems by introducing FedMobile (Liu et al., 2025a), a knowledge-contribution-aware federated

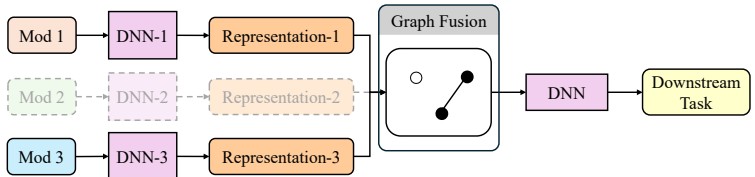

Figure 13: Illustration of graph fusion methods. We set modality 2 (white circle in the graph) as the missing modalities. Features from modality 1 and 3 can be aggregated via this graph fusion method and fused features maintains the same dimension.

learning framework that reconstructs missing features through cross-node multimodal knowledge transfer and robust aggregation.

Process-Based Distillation Methods focus on overall distillation strategies, like Mean Teacher Distillation (MTD) (Tarvainen & Valpola, 2017) and self-distillation (Zhang et al., 2019). These methods emphasize procedural learning over direct representation transfer. MTD enhances stability by using the exponential moving average of the student model's parameters as a teacher (figure 12a). Chen et al. (2024b) applied this to missing modality sentiment analysis, treating missing samples as augmented data. Li et al. (2022c) used MTD for Lidar-Radar segmentation, improving robustness against missing modalities.

Self-distillation helps a model improve by learning from its own soft representations (figure 12b). Wang et al. proposed the ShaSpec (Wang et al., 2023a), which utilizes distillation between modality-shared branches and modality-specific branches of available modalities. Based on ShaSpec, researchers proposed Meta-learned Cross-modal Knowledge Distillation (Wang et al., 2024a) to further weigh the importance of different available modalities to improve performance. Ramazanova et al. (2024) used mutual information and self-distillation for egocentric tasks, making predictions invariant to missing modalities. Shi et al. proposed PASSION (Shi et al., 2024), a self-distillation method designed to use the multi-modal branch to help other uni-modal branches of available modalities to improve multimodal medical segmentation performance with missing modality.

Hybrid Distillation Methods combine various distillation approaches to improve student performance. For medical segmentation, Yang et al. (2022) distilled teacher model knowledge (including logits and intermediate features) at every decoder layer, outperforming ACNet (Wang et al., 2021). Wang et al. (2023c) introduced ProtoKD, which captures inter-class feature relationships for improved segmentation under missing modality conditions. Recently, CorrKD (Li et al., 2024a) leverages contrastive distillation and prototype learning to enhance performance in uncertain missing modality cases.

Aforementioned methods address the missing modality problem and achieve good generalization during testing. However, except for some intermediate and self-distillation methods, most assume the complete dataset is available for training (means teacher can access full-modality samples during training), with missing modalities encountered only during testing. Therefore, most distillation methods are unsuitable for handling incomplete training datasets.

### 4.1.3 Graph Learning Based Methods

Graph-learning based methods leverage relationships between nodes and edges in graph-structured data for representation learning and prediction. We categorize approaches for addressing the missing modality problem into two main types: graph fusion and graph neural network (GNN) methods.

Graph Fusion Methods integrate multimodal data using a graph structure (figure 13), making them adaptable to various networks. For example, Angelou et al. (2019) proposed a method mapping each modality to a common space using graph techniques to preserve distances and internal structures. Chen & Zhang (2020) introduced HGMF, which builds complex relational networks using hyper edges that dynamically connect available modalities. Zhao et al. (2022) developed Modality-Adaptive Feature Interaction for Brain Tumor Segmentation, adjusting feature interactions across modalities based on binary existence codes. More recently, Yang et al. (2023a) proposed a graph attention based Fusion Block to adaptively fuse multimodal

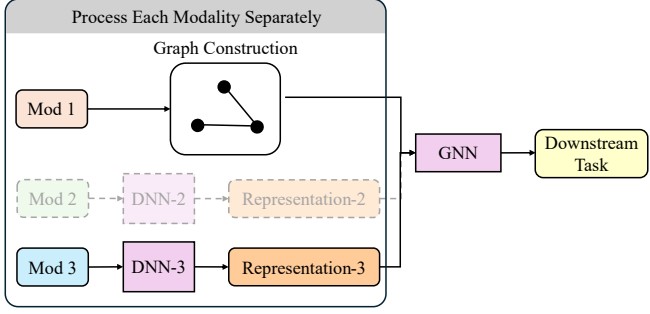

(a) Individual GNN Methods

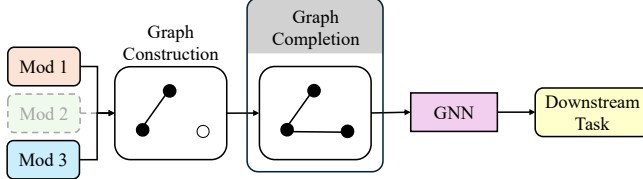

(b) Unified GNN Methods

Figure 14: Illustration of graph neural network (GNN) based methods. We set modality 2 as the missing modalities. (a) Each modality first passes through DNN or builds its graph, and then they are uniformly processed using GNNs. (b) This method first complete the graph from missing-modality samples via approaches like FeatProp (Malitesta et al., 2024), then process it by GNNs.

features, using attention-based message passing to share information between modalities. These fusion methods can be flexibly inserted into any network for integrating multiple modalities.

Graph Neural Network Methods directly encode multimodal information into a graph structure, with GNNs used to learn and fuse this information. Early approaches (Wang et al., 2015) employed Laplacian graphs to connect complete and incomplete samples. Individual GNN methods (figure 14a), such as DESAlign (Wang et al., 2024b), extract features using neural networks or GNNs and fuse them for prediction. Unified GNN methods ( figure 14b) first complete the graph and then use GNNs for prediction, such as in Zhang et al.'s M3Care (Zhang et al., 2022), which uses adaptive weights to integrate information from similar patients. Lian et al. (2023) proposed the Graph Completion Network, which reconstructs missing modalities by mapping features back to the input space. In recommendation systems, FeatPropMalitesta et al. (2024) propagates known multimodal features to infer missing ones, while MUSE (Wu et al., 2024b) represents patient-modality relationships as bipartite graphs and learns unified patient representations. In knowledge graphs, Chen et al. (2023b) introduced Entity-Level Modality Alignment to dynamically assign lower weights to missing/uncertain modalities, reducing the risk of misguidance in learning. Some work (Liu et al., 2020) addresses the missing-sensor (missing-modality) problem in topology-aware IoT systems by introducing a graph-based feature reconstruction module that uses neural message passing to recover features from available sensors. Some people Chen et al. (2023a) proposes a multi-graph convolutional framework for human activity recognition that explicitly handles asynchronous and incomplete IMU, magnetometer, and physiological signals, addressing missing-modality challenges in heterogeneous multi-sensor environments.

The methods above can leverage the graph structure to better capture relationships both within/between modalities and samples. However, these methods tend to have lower efficiency, scalability, and higher development complexity compared to other kind of approaches.

### 4.1.4 Multimodal Large Language Models (LLM)

The impressive transformative power of LLMs, like ChatGPT (Bubeck et al., 2023), can be explained by their impressive generalization capabilities across many tasks. However, our understanding of the world depends not only on language, but also on other data modalities, like vision and audio. This has led

researchers to explore MLLMs, designed to handle diverse user inputs across modalities, including cases with missing modalities, leveraging the flexibility of Transformers. Their architectures are similar to that shown in figure 10. Due to the idiosyncrasies of MLLMs, we distinguish them from normal transformer-based joint representation learning methods in section 4.1.1 and introduce them below. In some current MLLM architectures, LLMs act as feature processors, integrating feature tokens from different modality-specific encoders and passing the output to task-/modality-specific decoders. This enables LLM to not only capture rich inter-modal dependencies, but also naturally carry the ability to handle any number of modalities, that is, the ability to solve the missing modality problem.

Most MLLMs employ transformer-based modality encoders, such as CLIP, ImageBind (Girdhar et al., 2023), and LanguageBind (Zhu et al., 2023), which encode multimodal inputs into a unified representation space. Examples include BLIP-2 (Li et al., 2023b), which bridges the modality gap using a lightweight Querying Transformer, and LLaVA (Liu et al., 2024a), which enhances visual-language understanding with GPT-4-generated instruction data (Bubeck et al., 2023). These models are optimized for tasks like Visual Question Answering, Dialogue, and Captioning. Recent advancements extend output generation to multiple modalities, such as images. AnyGPT (Zhan et al., 2024) and NExT-GPT (Wu et al., 2023b) unify modalities like text, speech, and images using discrete representations and multimodal projection adaptors, enabling seamless multimodal interaction. CoDi (Tang et al., 2024) introduces Composable Multimodal Conditioning, allowing arbitrary modality generation through weighted feature summation. There are also some other methods that discard the modality encoders and use linear transformation directly, such as Fuyu (Bavishi et al., 2023) and OtterHD (Li et al., 2023a). Although MLLMs can flexibly handle any number of modalities, they have many disadvantages, such as the inconsistent multi-modal positional encoding, training difficulty, and high GPU resource requirements. Additionally, no specific MLLM benchmarks on missing modality problems have been proposed.

## 4.2 Model Combinations

Model combinations aim to use selected models for downstream tasks. These methods can be categorized into ensemble, dedicated training, and discrete scheduler methods. Ensemble methods combine predictions from multiple selected models through different types of aggregation methods, such as voting, weighted averaging, and similar approaches to improve the accuracy and stability. Dedicated training methods allocate different sub-tasks (e.g., different missing modality cases) to specialized individual models, focusing on specific sub-tasks or sub-datasets. In the discrete scheduler methods, users can use natural language instructions to enable the LLMs to autonomously select the appropriate model based on types of modalities and downstream tasks. We provide more details about these methods below.

### 4.2.1 Ensemble Methods

As detailed in the next paragraphs, ensemble learning methods allow flexibility in supporting different numbers of expert models to combine their predictions, as shown in figure 15. Multimodal Model Ensemble Methods: Early work (Wagner et al., 2011) in sentiment analysis employed ensemble learning to handle missing modalities, averaging predictions from uni-modal models. With deep learning advancements, the Ensemble-based Missing Modality Reconstruction network (Zeng et al., 2022) was introduced, leveraging weighted judgments from multiple full-modality models when generated missing modality features exhibit semantic inconsistencies. This type of multimodal model ensemble method, depicted in figure 15a, integrates various full-modality models to aid decision-making.

Unimodal Model Ensemble Methods: The general architecture of this method is shown in figure 15b, where each modality is processed by a uni-modal model, and only the available modalities contribute to decision-making. In multimodal medical image diagnosis, early studies found uniformly weighted methods performed better than weighted averaging and voting approaches (Yuan et al., 2012). In multimodal object detection, Chen et al. (2022) proposed a probabilistic ensemble method. This method does not require training and can flexibly handle missing modalities through probabilistic marginalization, demonstrating high efficiency in experiments. Li et al. (2023c) recently proposed Uni-Modal Ensemble with Missing Modality Adaptation, training models per modality and performing late-fusion training. Other approaches (Miech et al.,

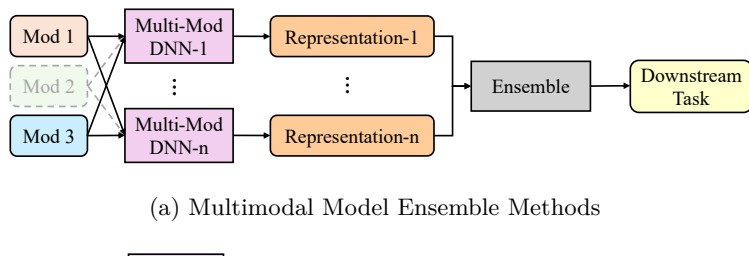

(a) Multimodal Model Ensemble Methods

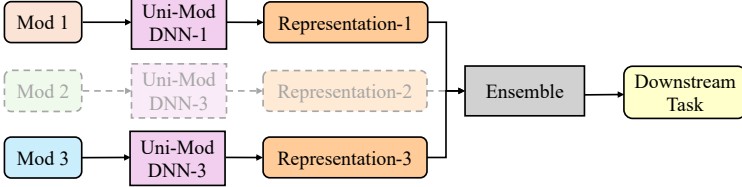

(b) Unimodal Model Ensemble Methods

Figure 15: Two general architectures for ensemble methods. (a) The multimodal model ensemble methods contain $n$ three-modal DNNs and produce final predictions based on the outputs of all $n$ DNNs. (b) Each modality is processed by a unimodal DNN in unimodal model ensemble methods, where final predictions are produced by aggregating the outputs of all accessible unimodal DNNs.

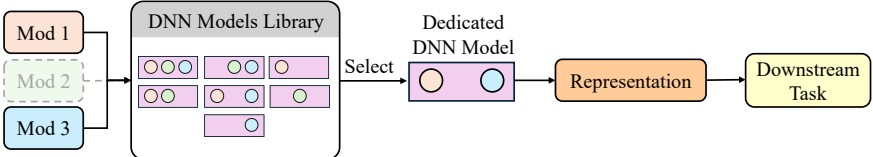

Figure 16: General idea of dedicated training methods. Assuming that modality 2 is missing, dedicated methods select the sample-suited model from DNN models library for training and testing. In the figure, the three modalities can form a total of seven models with different modality combinations. For easy understanding, we use colored circles in purple rectangles to represent the modalities that the model can handle. In order to adapt to the missing modality 2 (light green circles), dedicated methods usually select the DNN model (purple rectangle) that can handle modality 1 (light orange circles) and modality 3 (light blue circles) for training and testing.

2018) calculate feature-based weights to fuse modalities, where weights reflect feature importance for final predictions.

Hybird Methods: Dynamic Multimodal Fusion (DynMM) (Xue & Marculescu, 2023) uses gating mechanisms to select uni/multi-modal models dynamically. Recently, Cha et al. (2024) proposed Proximity-based Modality Ensemble (PME) to use a cross-attention mechanism with an attention bias to integrate box predictions from different modalities. PME can adaptively combine box features from uni-/multi-model models and reduce noise in multi-modal decoding.

### 4.2.2 Dedicated Training Methods

Dedicated training methods assign different tasks to specialized models. We show the general idea of those methods in figure 16. KDNet (Hu et al., 2020) was the first dedicated method proposed to handle different combinations of missing modalities. It treats uni-modality specific models as student models, learning multimodal knowledge from the features and logits of a multimodal teacher model. Those trained uni-modals can be used for different missing modality cases. Following KDNet, ACNet (Wang et al., 2021) also utilizes this distillation method but introduces adversarial co-training, further improving the performance. Lee et al. (2023b) proposed missing-modality-aware prompts to address missing modality problems based on the prompt learning by using input- and attention- level prompts for each kind of missing modality case.

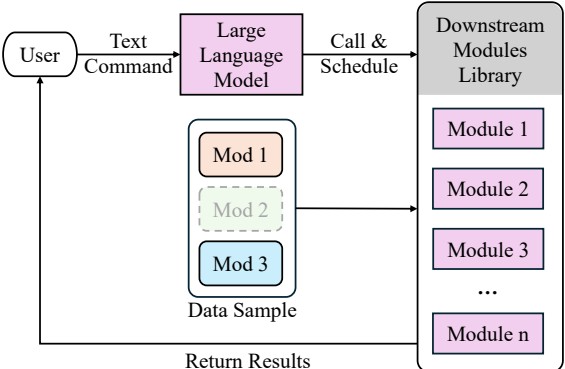

Figure 17: General procedure of discrete scheduler methods. The user types text instructions to make an LLM call and schedule the selected downstream task modules to process data samples and return results to the user. As long as there are sufficiently diverse downstream task modules, it is possible to handle the same downstream tasks with different numbers of modalities.

This method only needed 1% of the model parameters to be fine-tuned for downstream tasks. Similarly, some researchers (Reza et al., 2023) introduce different adapter layers for each missing modality case.

### 4.2.3 Discrete Scheduler Methods

In discrete scheduler methods (figure 17), LLMs act as controllers, determining the execution order of different discrete steps broken down from the major task/instruction. While the LLM does not directly process multimodal data, it interprets language instructions and orchestrates task execution across uni- and multi-modal modules. This structured yet flexible approach is particularly effective for outputs requiring sequential tasks, enabling the system to handle any number of modalities and naturally addressing missing modality problems. For example, Visual ChatGPT (Wu et al., 2023a) integrates multiple foundation models with ChatGPT to enable interaction through text or images, allowing users to pose complex visual questions, provide editing instructions, and receive feedback within a multi-step collaboration framework. Hugging-GPT (Shen et al., 2024) is an LLM-driven agent that manages and coordinates various AI models from Hugging Face. It leverages LLMs for task planning, model selection, and summarization to address complex multimodal tasks. ViperGPT (Surís et al., 2023) combines visual and language tasks into modular sub-programs, generating and executing Python code for complex visual queries without additional training to achieve effective outputs. There are other similar discrete scheduler approaches, such as MM-REACT (Yang et al., 2023b) and LLaVA-Plus (Liu et al., 2023c).

Some aforementioned dedicated and ensemble methods can flexibly handle the missing modality problem without additional training, but most of them require more model storage space, which is not feasible for many resource-constrained devices. For instance, DynMM requires storing various uni- and multi-modality models. As the number of modalities increases, the required number of models also rises. Also, unimodal model ensemble methods struggle to adequately model the complex inter-modality relationships to make final predictions. Although the some dedicated fine-tuning methods do not require too much time and consumption of resources such as GPU, it is still difficult to compare with full parameter training. In addition, discrete scheduler methods can solve a variety of tasks when there are sufficient types of downstream modules, but they usually require LLMs to respond quickly and understand human instructions accurately in real world scenarios.

## 5    Methodology Discussion

In the above section 3 and section 4, we divide the existing MLMM methods into four types from the aspects of data processing and strategy design: modality imputation, representation-focused, architecture-focused, and model combinations. We also further subdivide these four types into twelve categories, exploring a

Table 1: Comparison of deep multimodal learning with missing modality methods from four types and two aspects.

| Aspect | Type | Method | Pros and Cons |
|---|---|---|---|
| Data Processing | Modality Imputation | Modality Composition | Pros: Simple and effective for data augmentation. Cons: Unsuitable for pixel-level tasks (Composition); Modality number ↑, generative model number or training complexity ↑. |
| | | Modality Generation | |
| | Representation Focused | Coordinated-Representation | Pros: Flexible for novel modalities with better generalization. Cons: Hard to balance constraints and handle dataset imbalance. |
| | | Representation Composition | |
| | | Representation Generation | |
| Strategy Design | Architecture Focused | Attention-based | Pros: Attention methods scale well; Distillation and graph learning are effective. Cons: Attention methods often demand large computation & datasets; Distillation and graph learning struggle with incomplete datasets and training complexity. |
| | | Distillation-based | |
| | | Graph Learning-based | |
| | | Multimodal Large Language Model | |
| | Model Combinations | Ensemble | Pros: Effective for specific tasks. Cons: Grows complex with more modalities; Suffers from modality imbalance; Demands significant storage space. |
| | | Dedicated | |
| | | Discrete | |
| | | Scheduler | |

fine-grained methodology taxonomy proposed by us. Table 1 summarizes the overall pros and cons of these methods. Generative and distillation methods are the most common approaches; they are easy to implement and deliver strong performance. With the rise of Transformers, Transformers methods have become more popular due to their larger receptive fields and parallelism. However, indirect-to-task generation methods and most distillation methods (except for some intermediate (Zhang et al., 2023b) and self-distillation methods (Wang et al., 2023a; Shi et al., 2024)) are currently unable to handle incomplete training datasets (means cannot access missing modality data during training). Below we provide a concise analysis of these twelve methods from two aspects and four types.

## 5.1 Data Processing Aspect

**(1) Modality Imputation:** Modality composition methods operate directly on the input modality data level to address missing modality problems by combining existing data samples or filling missing data. However, they are typically not good at pixel-level downstream tasks and heavily rely on available modalities. On the other hand, modality generation methods employ generative models to synthesize missing modalities, mitigating the performance degradation caused by missing data. But these methods are often limited by the availability and number of full-modality samples and the increased training complexity due to the additional modalities. They also require extra storage for generative models.

**(2) Representation Focused:** Coordinated-representation methods allow novel modalities to be introduced by simply adding corresponding branches. However, as the number of constraints and modalities increases, balancing them becomes challenging. Correlation-driven methods often fail to handle missing modality samples in the test environment, as they are typically designed for pre-training with incomplete datasets. Similar to modality composition methods, representation composition methods attempt to recover missing modality representations by combining available-modality representations. Since representations typically carry more generalized information than modality data, these methods tend to yield better results. Representation generation methods further improve this process by generating representations based on relationships between modalities, allowing reconstruction of missing modality representations. Many studies have confirmed the effectiveness of this approach in handling missing modalities. However, if the dataset contains missing modality samples, indirect-to-task representation generation methods that aim to recover missing modality representations through reconstruction of modality data are not feasible. When the dataset has severely imbalanced modality combinations, generative methods may also fail, becoming overly reliant on existing modalities.

## 5.2 Strategy Design Aspect

**(1) Architecture Focused:** Some methods focus on designing model training or inference architecture to alleviate the performance degradation caused by missing modalities. Attention-based methods are valued by researchers because they can effectively capture the relationships between various modalities, are scalable in terms of dataset size, and are highly parallelizable. Single attention mechanisms used for modality fusion have the advantage of being plug-and-play. The drawbacks of transformer-based methods in this category are also relatively obvious: training multimodal transformers from scratch requires excessive training time and many GPU resources. Also, those attention-based methods generally require larger datasets due to their high model capacity and reliance on token-level interactions. While this enables stronger representation learning in large-scale settings, it also limits their applicability in data-scarce domains such as medical imaging. This trade-off highlights the need for data-efficient attention mechanisms or pretraining strategies in missing-modality scenarios. Recently popular PEL methods can effectively mitigate this drawback, but the generalization performance of this method is still not as well as full-parameter training or tuning. Although MLLMs can handle an arbitrary number of modalities with flexibility, they are constrained by training complexity and require substantial computational resources.

Distillation-based methods are relatively easy to implement, with student models learning how to reconstruct missing modality representations and inter-modal relationships from teacher models. Since the teacher model typically receives full-modality samples as input, it can simulate any missing modality cases during training, ensuring strong performance. However, most distillation-based methods are limited to the requirement of complete datasets and are inapplicable to datasets with inherent missing modalities. To our knowledge, only one intermediate distillation method (Zhang et al., 2023b) has attempted to input mismatched samples of the same class into the teacher and some self-distillation methods (Wang et al., 2023a; Shi et al., 2024) use representations of available modality branches for distillation purposes.

Graph learning-based methods capture intra- and inter-modal relationships more effectively, but as the number of modalities increases, the development complexity and inefficiency of these methods become more pronounced. They are also currently unsuitable for large-scale datasets.

**(2) Model Combinations:** These methods select models for prediction. Ensemble and dedicated training methods can be affected by imbalances in modality combinations within the dataset. As the number of modalities grows, the complexity and models to be trained also increase, especially for dedicated methods. Discrete scheduler methods requires an LLM to coordinate, but if callable modules are insufficient, downstream tasks may not be completed. Additionally, inference speed is limited by LLMs and modules. Model combinations methods also require significant model storage.

## 5.3 Recovery and Non-Recovery Methods & Some Statistics

Table 2: Another common taxonomy of recovery and non-recovery methods in multimodal learning is based on three stages: early, intermediate, and late stage, as seen in the conventional taxonomy. In this context, recovery and non-recovery methods refer to how missing modalities are handled at each stage.

| Methods | Early | Intermediate | Late | Hybrid |
|---|---|---|---|---|
| **Recovery** | Modality Composition (Yang et al., 2024) Modality Generation (Wang et al., 2024c) | Representation Composition (Wang et al., 2023b) Representation Generation (Woo et al., 2023) Distillation-based (Shen & Gao, 2019) Graph Learning-based (Malitesta et al., 2024) | Representation Generation (Ma et al., 2021c) Distillation-based (Li et al., 2022c) Coordinated-Representation (Ma et al., 2021b) | Distillation-based (Sikdar et al., 2023) |
| **Non-Recovery** | Dedicated (Wang et al., 2021) Discrete Scheduler (Wu et al., 2023a) | Representation Composition (Delgado-Escano et al., 2021) Attention-based (Ge et al., 2023) Graph Learning-based (Zhao et al., 2022) Multimodal Large Language Model (Wu et al., 2023b) | Coordinated-Representation (Liang et al., 2019) Distillation-based (Shi et al., 2024) Ensemble (Chen et al., 2022) | Not Applicable |

We have further classified MLMMs using the taxonomy in Table 1 to facilitate researchers to better distinguish MLMMs. This classification is based on the three-stage (early, intermediate, and late) classic multimodal learning taxonomy. Table 2 leverages this classic taxonomy of MLMMs and specifies whether they recover missing modalities or not. We also list reference papers for different methods. In our analysis of 315 papers, we found that 75.5% of the works focus on recovering missing modality information, while only 24.5% explore inference without modality recovery. Among the recovery methods, early-stage and intermediate-stage modality recovery methods account for 20.3% and 45.8%, respectively, with late-stage

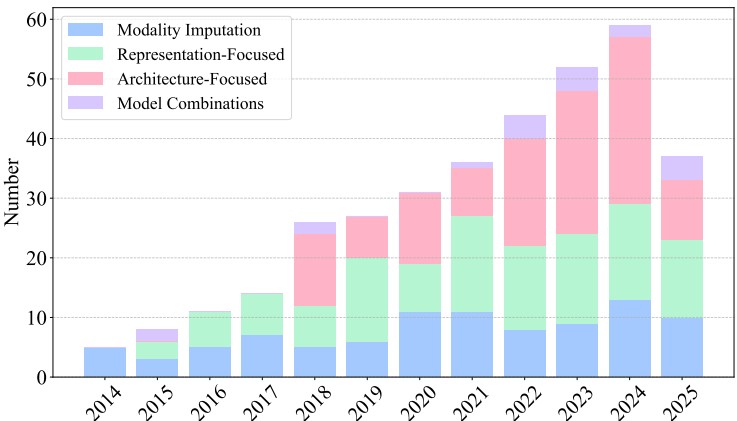

Figure 18: The annual statistics are based on our taxonomy of four methods: Modality Imputation, Representation-Focused, Architecture-Focused, and Model Combinations.

and multi-stage recovery methods accounting for 4.7%, each. For non-recovery methods, early, intermediate, and late-stage modality fusion methods represent 4.2%, 14.1%, and 6.3%, respectively. We were not able to find any non-recovery methods that combine multiple stages. We can observe that methods on recovering missing modality features in the intermediate stage account for the largest proportion. We think this is because recovering features, as opposed to the raw data, can avoid much noise and bias while providing more modality-specific/shared information. Additionally, compared to later-stage features, intermediate-stage features tend to be more enriched.

## 5.4 Technical Evolution Discussion

The development of MLMM methods has closely followed broader trends in machine learning and deep learning, with distinct methodological phases reflected in the taxonomy and publication statistics (2014–2025). Based on the papers we collected, we count the papers for each year according to the four major categories in our taxonomy mentioned above, and plotted them using a stacked bar chart (shown in figure 18).

Early research was dominated by data-driven heuristics and ensemble-style solutions, where missing modalities were addressed by retrieving similar samples from the dataset or by training separate unimodal models and combining them through ensemble strategies. This period corresponds to the initial emergence of Modality Imputation and Model Combinations, though the overall scale of research activity was still limited. With the rise of deep learning frameworks (e.g., PyTorch, TensorFlow) and the success of architectures such as ResNet, the community shifted its attention toward representation learning, leading to the rapid growth of Representation-Focused Models. From 2017 onward, this category became the most stable and persistent line of research, as learning aligned, shared, or generative representations enabled models to remain effective even when certain modalities were absent. At the same time, modality-level imputation methods evolved from simple sample-based matching to deep generative models, first through autoencoders and GANs and later through diffusion models, greatly improving the quality and generality of imputed modalities.

A notable transition occurred after 2018, possibly driven by the popularity of attention mechanisms, deep graph neural networks, and knowledge distillation. These techniques allowed models to dynamically adapt to arbitrary combinations of available modalities, leading to the rise of Architecture-Focused Models—a category that grows sharply in the statistics. These methods reframed missing modalities not merely as a data problem but increasingly as an architectural and training-strategy challenge, enabling flexible fusion, conditional routing, and intra/inter-modal knowledge transfer.

Most recently, the emergence of transformer-based foundation models has opened a new direction: applying large pretrained models to missing-modality conditions by using them as feature processors, task schedulers, or backbone architectures capable of consuming variable sets of modalities. This shift substantially expands

the design space of multimodal systems and suggests a future where missing-modality robustness is a built-in capability of general-purpose models rather than a specialized add-on.

Finally, although Model Combinations has historically remained a niche line of work, which we suspect is due to the cost of storing and maintaining multiple models, recent interest in Agentic AI systems may renew attention to this class of methods. As agents increasingly integrate multiple specialized tools, dynamic model selection and combination strategies may become more relevant in practical missing-modality scenarios. Together, these developments illustrate a clear technical evolution: from early heuristic data-level fixes to representation learning, to flexible architecture design, and finally toward foundation-model–based general multimodal intelligence. We note that applications and architectures are only weakly linked in this area, largely because modern deep learning architectures (e.g., attention) are highly versatile and transferable across tasks and applications. For instance, generative-adversarial-learning-based recovery methods are used in both breast-cancer prognosis (Arya & Saha, 2021) and visible–infrared person re-identification (Li et al., 2022d), and masked modeling strategy in human action recognition (Woo et al., 2023) closely parallel those used in brain-tumor segmentation (Liu et al., 2023a). As a result, once an architectural idea proves effective for handling missing modalities, it generalizes widely with minimal task-specific adaptation, which naturally weakens the direct coupling between applications and architecture.

**Limitations:** Due to the variety of settings in most missing-modality works, it is difficult to compare performances of different methods across datasets. This represents a research gap that we encourage the community to address.

## 6    Applications and Datasets

The collection of multimodal datasets is often labor-intensive and costly. In certain specific application directions, issues such as user privacy concerns, sensor malfunctions on data collection devices, and other factors can result in datasets with missing modalities. In severe cases, up to 90% of the samples may have missing modalities, making it challenging for conventional MLFM to achieve good performance. This has given rise to the task of MLMM. Since the factors causing incomplete datasets usually stem from different application directions, we introduce the following datasets based on the applications in MLMM tasks: Affective Computing, Medical Applications, Information Retrieval, Remote Sensing, Pervasive Computing, Robotic Vision and Sensing, and others. We categorize those datasets according to the applications and data type in Table 3. It should be noted that for the medical field, we only show part datasets due to their large number.

### 6.1    Affective Computing

Typically, the goal of affective computing is to classify the current emotional state by combining information from multiple modalities such as textual, auditory, and RGB information. This field has garnered significant attention due to its promising applications in various specific scenarios, including market research, health monitoring, and advertising. In the early stages of audio-visual affective computing research, researchers discovered that facial detection algorithm in data collection devices sometimes failed to capture faces (e.g., due to occlusion) or the recorded audio is too noisy to use (Cohen et al., 2004; Sebe et al., 2005), resulting in some samples that only contained audio or image. Consequently, researchers began exploring methods to enable affective computing models to continue functioning effectively when facing missing-modality samples. Currently, the datasets for applying affective computing can be roughly divided into two categories. One type of dataset  (Martin et al., 2006; Livingstone & Russo, 2018; Cao et al., 2014; Stappen et al., 2021a;b; Ringeval et al., 2013; Zheng & Lu, 2015; Zadeh et al., 2018; Busso et al., 2008; 2016; Poria et al., 2018; Yu et al., 2020; Wöllmer et al., 2013; Morency et al., 2011) involves using video, audio, text, and biosensors to determine human emotional states in real life or in movies. The other type (Guérin et al., 2013; Nguyen et al., 2018; Matsui et al., 2017) involves using text and images to assess emotional states in contexts such as comic books. Please refer to Table 3 for details.

Table 3: Common datasets used by deep MLMM methods divided by main applications and modality types. "Vision" includes data from visual sensors such as RGB images, Depth, Infrared, LiDAR, Radar, Event, and Optical Flow. "Bio-Sensors" includes data from sensors like: electrophysiological, respiratory sensors and so on. "Motion Sensors" includes data from sensors like accelerometers, gyroscopes, magnetometers, IMUs, barometric sensors, and gait sensors. "Physiological Signals" including ECG, PPG/BVP, EDA/GSR, RESP, EEG, EMG, EOG, skin temperature, and $SpO_2$. Other modalities include audio, text, CT scans, MRI, and skeleton.

| Applications | Modality Types | Common and Typical Datasets |
|---|---|---|
| **Affective Computing** | Vision+Audio | eNTERFACE'05 (Martin et al., 2006), RAVDESS (Livingstone & Russo, 2018), CREMA-D (Cao et al., 2014) |
| | Vision+Text | eBDtheque (Guérin et al., 2013), DCM (Nguyen et al., 2018), Manga 109 (Matsui et al., 2017) |
| | Vision+Bio-Sensors | Ulm-TSST (Stappen et al., 2021a;b), RECOLA (Ringeval et al., 2013), SEED (Zheng & Lu, 2015) |
| | Vision+Audio+Text | CMU-MOSI (Zadeh et al., 2016), CMU-MOSEI (Zadeh et al., 2018), IEMOCAP (Busso et al., 2008), MSP-IMPROV (Busso et al., 2016), MELD (Poria et al., 2018), CH-SIMI (Yu et al., 2020), ICT-MMMO (Wöllmer et al., 2013), YouTube (Morency et al., 2011) |
| **Medical Applications** | MRI, CT Scans | BRATS-series (Menze et al., 2014; Myronenko, 2019), ADNI (Jack Jr et al., 2008), IXI [https://brain-development.org/ixi-dataset] |
| | Bio-Sensors | StressID (Chaptoukaev et al., 2024), PhysioNet (Kemp et al., 2000), TCGA [https://portal.gdc.cancer.gov/], BCData (Huang et al., 2020), NuClick (Koohbanani et al., 2020), LYON19 (Swiderska-Chadaj et al., 2019) |
| | Electronic Health Record | MIMIC-CXR (Johnson et al., 2019), MIMIC-IV (Johnson et al., 2023), NCH (Lee et al., 2022), BCNB (Xu et al., 2021), ODIR (Li et al., 2021) |
| **Information Retrieval** | Vision+Text | Amazon Series Datasets (Lakkaraju et al., 2013; McAuley et al., 2015; He & McAuley, 2016; PromptCloud, 2017), MIR-Flickr25K (Duin), NUS-WIDE (Chua et al., July 8-10, 2009) |
| | Vision+Audio+Text | MSR-VTT (Xu et al., 2016), TikTok (Lin et al., 2023) |
| **Remote Sensing** | Vision | WHU-OPT-SAR (Li et al., 2022b), DFC2020 (Yokoya et al., 2020), MSAW (Shermeyer et al., 2020), Trento (Rasti et al., 2017), Houston (Piacentini, 2023a), Berlin (Hong et al., 2021), GRSS (Debes et al., 2014) |
| **Pervasive Computing** | Motion sensors | Opportunity (Roggen et al., 2010), RealDisp (Banos et al., 2014), DSADS (Barshan & Altun, 2010), PAMAP2 (Reiss, 2012), Shoaib (Shoaib et al., 2014), RealWorld-HAR (Sztyler & Stuckenschmidt, 2016), w-HAR (Burzer et al., 2025), Epilepsy (Villar et al., 2016), EMG Physical Action (Theodoridis, 2011), ERing (Wilhelm et al., 2015), and GENEActiv (Phillips et al., 2013) |
| | Physiological Signals | WESAD (Schmidt et al., 2018a), mBrain21 (Donckt, 2024), ETRI Lifelog (Oh et al., 2025) |
| | Electroencephalogram | Self RegulationSCP1 (Birbaumer et al., 1999), EEG-MMID (Schalk et al., 2004) |
| **Robotic Vision and Sensing** | Vision | RegDB (Ye et al., 2021), SYSU-MM01 (Wu et al., 2017), UWA3DII (Rahmani et al., 2016), Delivery (Zhang et al., 2023a), NuScenes (Caesar et al., 2020), MVSS (Ji et al., 2023) |
| | Vision+Audio | Ego4D-AR (Grauman et al., 2022), Epic-Sounds (Huh et al., 2023), Epic-Kitchens (Damen et al., 2022), Speaking Faces (John & Kawanishi, 2022) |
| | Vision+Skeleton/Inertial Data | MMG-Ego4D (Gong et al., 2023), Northwestern-UCLA (Wang et al., 2014), NTU60 (Shahroudy et al., 2016) |

## 6.2 Medical Applications

Medical domain requires comprehensive analysis from various modalities such as medical history, physical examination, imaging data, cell and gene data, which is exactly what multimodal learning is good at, many researchers developed multimodal intelligent systems. The current datasets for multimodal medical domain that contain data with missing modality problems are focused on below areas: Neuroimaging and Brain Disorders (Menze et al., 2014; Myronenko, 2019; Young et al., 2017; Jack Jr et al., 2008; Lesjak et al., 2018), Cardiovascular (Liu et al., 2016; Moody & Mark, 2001), Cancer Detection (Arya & Saha, 2020; Tomczak et al., 2015; Cheerla & Gevaert, 2019; Farooq et al., 2025), Women Health Analysis (Zhang et al., 2023b; Xu et al., 2021), Ophthalmology (Li et al., 2021; Zhang et al., 2022), Sleep Disorders (Lee et al., 2022; Marcus et al., 2013), Clinical Predictions (Johnson et al., 2016; 2023; 2019), Biomedical Analysis (Chaptoukaev et al., 2024; Schmidt et al., 2018b; Kemp et al., 2000), Computational Pathology Cui et al. (2022); Peng et al. (2024); Wang et al. (2025a); Xu et al. (2025), Radiology Chen et al. (2025); Cui et al. (2022); Liang et al. (2025), and Gene Schmauch et al. (2020) and Cell Ghahremani et al. (2022); Liu et al. (2025b); Palma et al. (2024); Tu et al. (2022) Representation learning. We have categorized some commonly used and representative datasets from list above by modality type, as shown in Table 3.

## 6.3 Information Retrieval

Information retrieval is a technology that automatically retrieves relevant content or data based on queries, historical behavior, preferences, and attribute data. It has received widespread attention due to its success on various platforms such as search engines and social media. It improves user satisfaction and information access experience through algorithmic analysis and prediction of content that users may find relevant. Multimodal learning makes modern retrieval systems possible because it can effectively process and analyze multiple types of information, including text, images, audio, etc. Due to user privacy concerns and data sparsity issues, some researchers have begun exploring MLMM approaches in retrieval systems. Many of these studies leverage datasets (Lakkaraju et al., 2013; McAuley et al., 2015; He & McAuley, 2016; PromptCloud, 2017; Duin; Chua et al., July 8-10, 2009; Xu et al., 2016; Lin et al., 2023) from websites such as Amazon and TikTok for their research. We list those datasets based on modality type in Table 3.

### 6.4 Remote Sensing

By leveraging multimodal learning, we can integrate various types of visual data, such as Synthetic Aperture Radar data and multi-/hyper-spectral data. As a result, various datasets have emerged (Yokoya et al., 2020; Shermeyer et al., 2020; Rasti et al., 2017; Piacentini, 2023a; Hong et al., 2021; Debes et al., 2014). Those datasets enable the assessment and analysis of different environmental conditions, resources, and disasters on earth from satellites or aircraft. Such capabilities can significantly aid in environmental protection, resource management, and disaster response. In practice, remote sensing tasks, such as multimodal land cover classification tasks, optical modalities may be unavailable due to cloud cover or sensor damage. Therefore, it is also necessary to address the missing modality problem for remote sensing problems.

### 6.5 Pervasive Computing

The pervasive computing community has also developed a wide range of multimodal sensing datasets that capture the complexity of real-world mobile, wearable, and IoT environments. They can be meaningfully organized by task category and reflect the dominant research problems across pervasive and mobile computing. (1) Human Activity Recognition. A broad class of datasets focuses on fine-grained activity recognition, gesture understanding, and daily-living behavior modeling. Representative collections include Opportunity (Roggen et al., 2010), RealDisp (Banos et al., 2014), DSADS (Barshan & Altun, 2010), PAMAP2 (Reiss, 2012), Shoaib (Shoaib et al., 2014), RealWorld-HAR (Sztyler & Stuckenschmidt, 2016), w-HAR (Burzer et al., 2025), ERing (Wilhelm et al., 2015), and EMG Physical Action (Theodoridis, 2011). These datasets capture heterogeneous, body-mounted signals—accelerometry, gyroscope, magnetometer, EMG—collected in naturalistic environments that inherently involve sensor dropout, misalignment, and variable sampling. (2) Physiological Monitoring. Many datasets target long-term physiological monitoring. Examples include WESAD (Schmidt et al., 2018a),Epilepsy (Villar et al., 2016), and combining biosignals such as ECG, EDA, BVP, RESP, or skin temperature, with motion sensors or contextual smartphone streams. Their free-living collection protocol results in multimodal recordings with non-wear intervals, irregular sampling, and missing physiological channels. (3) Cognitive State Inference. Datasets for cognitive-state modeling, mental workload analysis, and motor imagery—such as Self RegulationSCP1 (Birbaumer et al., 1999) and EEG-MMID (Schalk et al., 2004)—contain high-dimensional EEG signals. EEG's susceptibility to noise, channel corruption, and abrupt signal loss naturally produces structural missingness at both temporal and spatial (channel) levels. Taken together, these task-driven dataset categories illustrate that pervasive computing data are inherently noisy, asynchronous, and incomplete, with many recordings containing long gaps, irregular sampling, and sensor-level failures. As a result, they serve not only as benchmarks for multimodal recognition, context understanding, and mobile health monitoring, but also as natural testbeds for evaluating MLMM methods under real-world structural missingness. Their modality heterogeneity, device-specific characteristics, and episodic sensor dropout further make them valuable for assessing robustness, cross-device generalization, and lightweight recovery strategies necessary for deployment on resource-constrained edge platforms. The classification of above dataset on modalities can be found in Table 3.

### 6.6 Robotic Vision and Sensing

Robotic vision is the field that enables robots to perceive and understand their environment through visual sensors by acquiring and processing image data. This includes tasks such as object and facial recognition, environment modeling, navigation, and behavior understanding, allowing robots to autonomously perform complex operations and interact with humans. Typically, the modality combination formula in robotic vision tasks can be expressed as RGB+X, where X can include, but are not limited to, LiDAR, radar, infrared sensors, depth sensors and auditory sensors. Both RGB and other sensor data can be missing for various reasons. Below, we introduce five common tasks in robotic vision concerning the missing modality problems. (1) Multimodal Segmentation: This task aims to input multimodal data and use segmentation masks to locate objects of interest. In the context of missing modalities, this task is common in datasets for autonomous vehicles, where sensors might fail due to damage or adverse weather conditions. Typical data combinations include RGB+(Depth/Optical Flow/LiDAR/Radar/Infrared/Event data) (Takumi et al., 2017; Liu et al., 2023b; Caesar et al., 2020; Zhang et al., 2023a). Other datasets focus on indoor

scene segmentation (Silberman et al., 2012; Ha et al., 2017) (RGB, Depth, Thermal) and material segmentation (Liang et al., 2022) (RGB, Angle of Linear Polarization, Degree of Linear Polarization, Near-Infrared images). (2) Multimodal Detection: Similar to multimodal segmentation, multimodal detection aims to locate objects of interest using bounding boxes. Commonly used multimodal detection combinations in the context of missing modalities include: RGB with Depth (Silberman et al., 2012; Lai et al., 2011) and RGB with Thermal (Hwang et al., 2015; kaggle, 2020). (3) Multimodal Activity Recognition: This task can input data from the aforementioned visual sensors and also use audio cues to recognize human activities. Common datasets (Grauman et al., 2022; Huh et al., 2023; Damen et al., 2022; Gong et al., 2023; Wang et al., 2014; Shahroudy et al., 2016) for this task are combinations of RGB with Audio, Depth, Thermal, Inertial and skeleton data. (4) Multimodal Person Re-Identification: This task aims to use depth and thermal (Ye et al., 2021; Wu et al., 2017) information to robustly identify individuals under varying lighting conditions, such as at night. (5) Multimodal Face Anti-Spoofing: This task utilizes multiple sensors (such as RGB, infrared, and depth cameras) to detect and prevent spoofing in facial recognition systems (John & Kawanishi, 2022; Zhang et al., 2020; Liu et al., 2021a). Its purpose is to enhance the system's ability to identify fake faces (such as photos, videos, masks, etc.) by comprehensively analyzing different modalities, thereby increasing the security and reliability of facial recognition systems.

**Other Applications:** There are many other areas where MLMM methods are being explored. For example, conventional audio-visual classification (Vielzeuf et al., 2018); multimodal large language models in visual-dialogue (Bubeck et al., 2023); audio-visual question answering (Park et al., 2024); captioning (Liu et al., 2023c); hand pose estimation using depth images and heat distributions (Tompson et al., 2014; Choi et al., 2017); knowledge graph completion utilizing multimodal data with missing modalities (Liu et al., 2019); multimodal time series prediction, such as stock prediction (Luo et al., 2024) and air quality forecasting (Zheng et al., 2015); multimodal gesture generation on how to create natural animations (Liu et al., 2022); multimodal analysis of single-cell data (Mimitou et al., 2021) in biology.

**Discussion:** Current MLMM research primarily focuses on the areas of affective computing, medical applications, remote sensing, information retrieval, pervasive computing and robotic vision and sensing. Among these, a significant portion of research is dedicated to addressing the missing modality problem in RGB, text, and audio-based sentiment analysis, MRI segmentation and clinical prediction, and multi-sensor segmentation for autonomous vehicles. In contrast, research efforts in MLMM for streaming data and scientific fields are relatively limited. In addition, the majority of the publicly available datasets mentioned above are complete in terms of modalities, and naturally occurring datasets with missing modalities are rare. As a result, these tasks are often evaluated by considering all possible combinations of missing modalities based on the existing types of modalities in the dataset, and then performing training and testing on the performance of the missing modalities, supplemented by different missing modality rates. Perhaps more future work could try to emulate Pervasive Computing by collecting multimodal datasets with real-world modal data from different models, including those with real sensor damage or different sampling rates of time-series information, for model validation. In the works we reviewed, 38% controlled for varying degrees of missing modality rates and tested accordingly, while the remaining 62% used random modality missing rates for training and testing. When categorizing missing modality rates into mild ($<30\%$), moderate ($30\%$-$70\%$), and severe ($>70\%$) levels, the proportions are 72.6%, 82.2%, and 69.9%, respectively. It is worth noting that a single study may perform validation across multiple missing modality rate levels. Additionally, we found that 36.4% of the works trained on incomplete training datasets (cannot access missing modality data during training), while 63.6% actually focused on testing with missing modalities. We can see that there is still small number of research on incomplete training datasets, so it is important to emphasise the need for more research to this field.

Beyond the above observations, we also note several underexplored yet crucial research gaps. First, current benchmarks rarely model realistic missingness patterns such as systematic or asynchronous sensor failures, despite their prevalence in pervasive and wearable computing. For example, in wearable activity recognition (Nweke et al., 2018), IMUs and physiological sensors frequently sample asynchronously or drop entire segments, producing structured missingness that cannot be captured by random masking. Second, the semantic importance of different modalities is typically assumed to be uniform, and there is no principled framework for measuring modality contribution or uncertainty under missing conditions. Third, evalua-

tion protocols remain highly inconsistent across studies, with varying definitions of missing rates, train–test configurations, and dataset preprocessing. For example, SMIL (Ma et al., 2021c) considers 90% of the multimodal samples in the training set with missing modalities, but DrFuse (Yao et al., 2024) only considers the case of random missing modalities and does not consider the performance at different missing modality rates. Fourth, most existing models struggle to generalize across domains, devices, or spatiotemporal distributions, highlighting the need for scalable and transferable foundations for MLMM. Finally, there is a lack of rigorous real-world case-study evaluations and systematic testing on out-of-distribution multimodal datasets, both of which are essential for assessing robustness under practical deployment conditions. Addressing these gaps will be critical for developing MLMM systems that can reliably operate in real-world multimodal environments.

## 7 Open Issues and Future Research Directions

### 7.1 Accurate Missing Modality/Representation Data Generation

Some researchers (Ma et al., 2021c) have shown that recovering, reconstructing, or generating missing modalities or their intermediate representations can improve model robustness when encountering incomplete multimodal inputs. However, the generated content often contains artifacts or hallucinations, stemming either from the generative models themselves or from imperfections and biases in their training data. Beyond improving generative fidelity, a key future direction is to better understand informational complementarity and redundancy across modalities: not all modalities contribute equally to downstream tasks, and some provide overlapping signals that can be safely inferred, while others contain unique information that is difficult to reconstruct without introducing bias. Therefore, exploring principled ways to generate accurate, unbiased, and task-relevant missing modalities—while leveraging modality complementarity and avoiding over-reliance on redundant cues—will be an important avenue for future research.

### 7.2 Recovery or Non-Recovery Methods ?

In MLMM, *"to recover or not to recover the missing modality"*, that is the question. According to our observations, most current approaches to handling MLMM can be broadly categorized into two types: one focusing on recovering the missing modal information so that the multimodal model can continue functioning (Wang et al., 2023a; Ma et al., 2021c), and the other attempting to make predictions using only the data modalities available (Hu et al., 2020; Zhang et al., 2024). However, some studies (Yao et al., 2024) have shown that the recovered modality information might be ignored or dominated by the existing modality information due to the imbalance missing rates of different modalities in the training set, leading the model to still rely on the available modalities when making predictions. Moreover, Lin et al. (2024) have found that in situations where crucial modality information is missing, the model might be unduly influenced by less important existing modality information, resulting in incorrect outcomes. In such cases, attempting to recover missing modality information can potentially alleviate this problem. Consequently, determining under what circumstances recovery or non-recovery methods are more beneficial for model predictions, as well as how to measure whether the recovered modality information is not dominated by other modality information or is unbiased, remains a significant challenge. Addressing these issues is essential for advancing the effectiveness and reliability of deep multimodal learning with missing modality methods.

### 7.3 Benchmarking and Evaluations for Missing Modality Problems

Benchmarking and fair evaluation play crucial roles in guiding the field. Many works are trained/tested under different missing modality settings, which makes it difficult for researchers to find works with the same settings for comparison. Also, the recent rise of large pre-trained models, such as GPT-4, has given researchers hope for achieving Artificial General Intelligence. Consequently, more large models (Liu et al., 2024a) are integrating visual, auditory, and other modal information to realize the vision of Multimodal Large Language Models. It is important to note that recent multimodal models (especially transformer-based models) are typically trained on significantly larger and more diverse datasets than earlier approaches. Early studies on missing modality learning often relied on constrained, domain-specific datasets (e.g., medical

imaging dataset) to validate model architectures and theoretical assumptions. In contrast, recent works leverage large-scale multimodal pretraining, sometimes involving billions of image-text or audio-visual pairs collected from the web. This expansion in data scale not only improves model generalization but also changes the nature of the learning problem, thus some recent models can implicitly infer missing information from learned priors rather than explicit modality alignment. Therefore, when comparing results across methods, it is crucial to account for these disparities in dataset scale, diversity, and accessibility, as they directly affect both performance and reproducibility. Although some research has been done based on those models, detailed analysis and benchmarks are urgently needed to evaluate the performance of MLLMs on missing modality problems. We call on the community to build a work similar to MultiBench (Liang et al., 2021), covering common datasets mentioned in section 6 and settings of missing modality problems, to help researchers conduct more systematic research.

### 7.4 Method Efficiency

Current MLMM methods often overlook the need for more lightweight and efficient methods. For instance, some model combinations methods (Hu et al., 2020; Xue & Marculescu, 2023) require training an independent model for each modality combination. Similarly, methods (Chen et al., 2024a; Zhang et al., 2023c) that aim to recover missing modality information typically involve using a distinct model for each kind of missing information or a large unified model to generate all modalities. Although these approaches generally perform well, they are often too heavy. In the real world, many multimodal models need to be deployed on resource-constrained devices, such as space or disaster-response robots. These devices cannot accommodate high-performance GPUs and are prone to sensor damage, which can be difficult/costly to repair. Therefore, there is an urgent need to explore efficient and lightweight solutions for MLMM that can operate effectively under these constraints.

### 7.5 Multimodal Streaming/Temporal Data with Missing Modality

Currently, only a small portion of research focuses on handling missing modalities in multimodal streaming or temporal data. An example is sentiment analysis (Lin & Hu, 2023). Long sequences of multimodal data—such as RGB+X video streams and multimodal time-series signals—are common in real-world settings and are essential for tasks like anomaly detection and video tracking, especially in robotic systems and pervasive computing. In such scenarios, missing modalities often occur asynchronously, creating not only spatial information gaps but also temporal misalignment between surviving modalities. This misalignment makes recovery, fusion, and prediction more challenging, as models must infer missing information while maintaining consistent temporal dependencies. Therefore, MLMM needs to address both missing-modality problems and temporal alignment issues in streaming data to advance this field further.

### 7.6 Multimodal Reinforcement Learning with Missing Modality

Multimodal reinforcement learning leverages information from different sensory modalities, enabling agents to learn effective strategies in environments. This approach has wide applications in areas such as robotic grasping, drone control, and driving decisions. However, these agent-based tasks often face practical challenges, as real-world scenarios frequently have sensor failures or restricted access to certain modality data, compromising the robustness of the algorithms. Currently, only a limited number of papers (Vasco et al., 2021; Lee et al., 2021; Awan et al., 2022) aim to address the problem of missing data or modalities in reinforcement learning. Therefore, we hope the community can focus more on the practically significant missing modality problem in reinforcement learning.

### 7.7 Multimodal AI with Missing Modality for Natural Science

In scientific fields such as drug prediction (Deng et al., 2020) and materials science (Muroga et al., 2023), multimodal learning plays a key role by integrating diverse data types like molecular structures, genomic sequences, and spectral images. This integration can enhance predictive accuracy and uncover new insights. However, implementing multimodal learning in these domains is challenging due to data access restrictions,

high acquisition costs, and the incompatibility of heterogeneous data. These issues often result in missing modalities, hindering the potential of multimodal models. Despite the critical need, there has been limited research (He et al., 2024) on MLMM in scientific applications. Addressing this gap is essential for advancing AI-driven discoveries, requiring new methods to handle incomplete multimodal datasets and create robust models capable of learning despite missing data.

### 7.8 Handling Practical Missing Modality Problems in Real-World

A promising yet still insufficiently explored direction for MLMM lies in real world scenarios—including IoT infrastructures, wearables, smart homes, mobile sensing, and distributed environmental sensor networks. Although several studies in pervasive computing have begun to address missing-modality challenges in these settings, current efforts remain fragmented and lack holistic evaluation. These environments constitute one of the most consequential real-world deployment contexts for multimodal models, where missingness is not occasional or synthetic but structural and persistent due to heterogeneous devices operating under constraints such as limited battery life, bandwidth restrictions, hardware degradation, asynchronous sampling, and intermittent connectivity. As a result, multimodal streams often exhibit long gaps, irregular intervals, and complete sensor dropout—conditions rarely captured by existing benchmarks. Empirical findings from mobile and wearable sensing further highlight the gap: human activity recognition systems frequently encounter inconsistent sampling, device failures, and user-dependent variability (Nweke et al., 2018); wrist-worn sensing studies report prolonged non-wear periods and dropout of inertial or physiological channels (Van Der Donckt et al., 2024); and systems integrating IMUs, magnetometers, and physiological sensors exhibit structural missingness induced by asynchronous fusion (Chen et al., 2023a). While these studies provide valuable insights, MLMM still lacks general frameworks capable of handling long-term partial observability, heterogeneous devices, asynchronous and irregular multimodal streams, and multi-agent inconsistencies in a unified manner. More systematic evaluations and standardized benchmarks are urgently needed. A complementary future direction involves developing resource-aware MLMM robustness strategies. Although prior work has explored lightweight methods such as signal augmentation and cross-modality distillation (Jeon et al., 2021), and energy-efficient imputation pipelines tailored for mobile health applications (Hussein et al., 2024b), these techniques remain application-specific and do not yet form a generalizable toolkit. Likewise, generative approaches such as SensorGAN (Hussein & Bhat, 2024) show promise for reconstructing missing sensor channels, but broader assessments of their scalability, latency, and reliability in real-world environments are still lacking. Overall, handling practical missing modality problems in real world presents a critical frontier for MLMM research. Despite early progress, the field still needs more comprehensive evaluations, standardized benchmarks, and general-purpose robustness principles. Ultimately, enabling multimodal AI systems that remain reliable under persistent sensor failures and dynamically evolving, heterogeneous environments will require unified methodologies and architectures explicitly designed for real-world sensing constraints.

## 8 Conclusion

In this survey, we present the first comprehensive survey of Deep Multimodal Learning with Missing Modality. We begin with a brief introduction to the motivation of the missing modality problem and the real-world reasons that underscore its significance. Following this, we summarize the current advances based on our fine-grained taxonomy and review the applications and relevant datasets. Finally, we discuss the existing challenges and potential future directions in this field. Although more and more researchers are involved in studying the problem of missing modality, we are also concerned about some urgent issues that need to be addressed, such as a unified benchmark for testing and the need for a wider range of applications With our comprehensive and detailed approach, we hope that this survey will inspire more researchers to explore missing modality problems.

**Limitations:** Because our survey aims to cover deep multimodal learning with missing modalities across as many domains as possible from 2012 to 2025, our taxonomy focuses on the practical machine learning perspective rather than analysing methods in a theoretical way. Moreover, although we separately review how large models address missing modalities, the rapid development of foundation model ecosystems suggests

that this taxonomy may require further refinement as new architectures, training paradigms, and modality agnostic interfaces continue to emerge.

In addition, method efficiency remains an underexplored yet increasingly critical challenge. Currently, most existing MLMM approaches rely on computationally heavy architectures, expensive cross modal alignment modules, or complex imputation pipelines. Their scalability to edge devices, resource constrained deployments, and long horizon streaming scenarios is still limited. Future research should systematically examine efficiency trade offs, including latency, memory footprint, modality specific bottlenecks, and asymmetry between training and inference, to enable practical and deployable MLMM systems.

Additionally, current benchmarks rarely capture realistic missingness patterns such as systematic, asynchronous, intermittent, or catastrophic sensor failures. Existing studies also tend to assume uniform modality importance and lack principled frameworks for quantifying modality contribution, uncertainty, or redundancy under missing conditions. Evaluation protocols remain inconsistent across works, hindering fair comparison. Beyond this, most methods still struggle to generalize across domains, devices, and spatiotemporal distributions.

Finally, while MLMM research is most active in robotic vision and sensing, medical applications, affective computing, remote sensing, and pervasive computing, its application to real-world sensing systems and natural science domains such as IoT infrastructures, wearables, smart homes, mobile sensing, distributed environmental sensor network, climate modeling, earth observation, molecular biology, and materials discovery remains sparse. These fields feature rich, heterogeneous, and often severely incomplete modalities, for example missing experimental measurements or partially observed molecular conformations. Extending MLMM research to these areas will require domain aware missingness models, physics and biology informed constraints, and better theoretical understanding.

Addressing these limitations, along with developing stronger theoretical frameworks, more realistic benchmarks, efficiency aware architectures, and broader cross domain applicability, will help guide the design of more robust, transferable, and scientifically grounded future MLMM research.

## Acknowledgments

We appreciate all the valuable feedback from the anonymous reviewers and the TMLR editors. In addition, Renjie would like to thank all co-authors for their continued support.

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
