# Supplementary Material for "Deep Multimodal Learning with Missing Modality: A Survey"

**Renjie Wu**                                                         *renjie.wu@anu.edu.au*
*The Australian National University*

**Hu Wang**                                                          *hu.wang@mbzuai.ac.ae*
*Mohamed bin Zayed University of Artificial Intelligence*

**Hsiang-Ting Chen**                                              *tim.chen@adelaide.edu.au*
*Adelaide University*

**Gustavo Carneiro**                                             *g.carneiro@surrey.ac.uk*
*The University of Surrey*

**Reviewed on OpenReview:** *https://openreview.net/forum?id=tc7RFcx4hT*

## A  Paper Collection

The following shows the full names of the mentioned conference and journal abbreviations in the "Paper Collection" paragraph of Section Introduction. The collected papers come from, but are not limited to, the following conferences (such as the Association for the Advancement of Artificial Intelligence (AAAI), International Joint Conference on Artificial Intelligence (IJCAI), Conference on Neural Information Processing Systems (NeurIPS), International Conference on Learning Representations (ICLR), International Conference on Machine Learning (ICML), IEEE Conference on Computer Vision and Pattern Recognition (CVPR), International Conference on Computer Vision (ICCV), European Conference on Computer Vision (ECCV), Annual Meeting of the Association for Computational Linguistics (ACL), Conference on Empirical Methods in Natural Language Processing (EMNLP), ACM SIGKDD Conference on Knowledge Discovery and Data Mining (KDD), ACM International Conference on Multimedia (ACM MM), International Conference on Medical Image Computing and Computer-Assisted Intervention (MICCAI), etc.) and journals (such as IEEE Transactions on Pattern Analysis and Machine Intelligence (TPAMI), IEEE Transactions on Image Processing (TIP), IEEE Transactions on Medical Imaging (TMI), IEEE Transactions on Multimedia (TMM), Journal of Machine Learning Research (JMLR), TMLR (Transactions on Machine Learning Research), etc.).