# OpenReview forum: "Deep Multimodal Learning with Missing Modality: A Survey"
_TMLR — Accepted by TMLR_

### Review · Reviewer_PJyo · 2025-10-28

**Summary Of Contributions:**

The manuscript is a survey of Deep Multimodal Learning with Missing Modality (MLMM). It proposes a two-axis taxonomy: Data Processing and Strategy Design. And it expands these into 12 concrete categories (e.g., modality/representation composition vs. generation; attention-, distillation-, and graph-based methods; ensembles, dedicated training, discrete schedulers; plus a separate bucket for MLLMs). It compiles 354 papers (2012–Aug 2025), outlines trends, catalogs applications and datasets, and concludes with open problems. It also offers clear visualizations of the different categories, which I found useful for understanding the main ideas behind the methods in each category. The text flows naturally and is well written, with only a couple of places that are harder to read. Overall, I found this manuscript useful for readers (potentially new to this field) to get a big picture of different methodologies.

**Additional Comments:**

It was an interesting read! Good job!

**Audience:**

Yes

**Audience Explanation:**

The manuscript covers an important problem, and it is useful for people interested in ML in general and those who specialize in ML applications.

**Claims And Evidence:**

Yes

**Claims Explanation:**

Yes, the manuscript supports the main claims and lists the papers. However, due to the scale of the papers published in the field, it is possible that they missed some papers. This is because I came across some papers in computational pathology that were not included and I list a few of them in the next sections.

**Requested Changes:**

I have come across multiple papers in the space of computational pathology that the survey doesn't cover, and I believe there are even more. So, I'd strongly recommend reevaluating the paper inclusion scheme for the medical domain to cover fields such as medical imaging, gene representation learning, cell representation learning, and radiology and pathology, as work in these often associate with missing modality:
1. Distilled Prompt Learning for Incomplete Multimodal Survival Prediction, CVPR 2025 [distilled models category]
2. A deep learning model to predict RNA-Seq expression of tumours from whole slide images, Nature Comm 2020 [modality generation]
3. Deep Learning-Inferred Multiplex ImmunoFluorescence for Immunohistochemical Image Quantification, Nature Mac Intl 2022 [modality generation]

The other point that I would like the author to clarify, especially for the attention network, is that these methods are generally more data hungry than other older methods. However, it has not been covered in the draft. Also, chronologically speaking, newer methods tend to train and scale with more data in their method (potentially a main reason for better generalization), which makes the head-to-head comparison of different methods even harder. So, I'd like the author to comment on this as well.

There are some editorial comments:
1. I found this paragraph hard to read and would recommend rewriting it for better clarity: "In addition, indirect-to-task representation generation methods can access **the missing modality data of the missing modality samples during training, which means that the training dataset is actually complete**, so this type of method usually performs better than direct-to-task representation generation methods." To be specific, I highlighted the part that is confusing to me.

2. I found this "self-distillationZhang et al. (2019)." which just needs an extra edit midway in the sentence ;)

---

> ### Author Response · Authors · 2025-11-21
>
> Sorry for the late rebuttal. We sincerely thank the reviewer for the thorough evaluation and constructive suggestions. Below, we provide detailed responses to all questions and requests. Furthermore, we have revised the manuscript in our new submission, and all modifications have been highlighted for ease of reference.
>
> ---
>
> ## **Q: Missing related works of computational pathology, gene/cell representation learning, and radiology.**
>
> **A:**
> We thank the reviewer for highlighting this important omission.
>
> **Action:**
>
> * We have revised the paper inclusion scheme for biomedical domains and pathology:
>
>   * Included the three representative works mentioned by the reviewer and an additional ten representative papers from computational pathology, medical imaging, and related biomedical areas.
>   * Added corresponding datasets used in the newly added papers to Table 3.
>   * Edited Section 6.2 to discuss representative applications in gene/cell representation learning, radiology, and pathology.
>   * Integrated the missing methods into the relevant method sections throughout the survey.
> * All updates are highlighted in our revised submission.
>
> ---
>
> ## **Q: Data Requirements of Attention-Based Methods**
>
> **A:**
> We appreciate the reviewer’s insightful comment. Attention-based models generally require larger datasets due to their high parameterization and dependency on extensive feature interactions. This characteristic can make them less suitable in data-scarce scenarios compared to older or lighter-weight architectures.
>
> **Action:**
>
> * Added a short discussion in Section 5.2 “Strategy Design Aspect” to highlight the data-hungry nature of attention-based methods and their limitations in missing-modality settings, especially in medical and low-resource domains.
>
> * The added text is:
>
>   > “Also, those attention-based methods generally require larger datasets due to their high model capacity and reliance on token-level interactions. While this enables stronger representation learning in large-scale settings, it also limits their applicability in data-scarce domains such as medical imaging. This trade-off highlights the need for data-efficient attention mechanisms or pretraining strategies in missing-modality scenarios.”
>
> * Updated the “Cons” of attention-based methods in Table 1.
>
> ---
>
> ## **Q: Clarification of the difference in training data requirements between the new and old methods**
>
> **A:**
> We agree with the reviewer that newer methods tend to rely on much larger datasets, often scaling to web-scale multimodal corpora. This trend complicates direct comparison with earlier approaches that were trained on smaller or domain-specific datasets.
>
> **Action:**
>
> * Added a paragraph in Section 7.3 “Benchmarking and Evaluations for Missing Modality Problems” to clarify temporal differences in data scales and caution against direct comparison.
> * The added text is:
>
>   > “It is important to note that recent multimodal models (especially transformer-based models) are typically trained on significantly larger and more diverse datasets than earlier approaches. Early studies on missing modality learning often relied on constrained, domain-specific datasets (e.g., medical imaging dataset) to validate model architectures and theoretical assumptions. In contrast, recent works leverage large-scale multimodal pretraining, sometimes involving billions of image-text or audio-visual pairs collected from the web. This expansion in data scale not only improves model generalization but also changes the nature of the learning problem, thus some recent models can implicitly infer missing information from learned priors rather than explicit modality alignment. Therefore, when comparing results across methods, it is crucial to account for these disparities in dataset scale, diversity, and accessibility, as they directly affect both performance and reproducibility.”
>
> ---
>
> ## **Q: Rewriting the description of indirect-to-task representation generation methods**
>
> **A:**
> We appreciate the reviewer’s suggestion for improving clarity. Our intended meaning was that these methods have access to complete multimodal training samples, enabling them to simulate missing-modality cases via modality dropping.
>
> **Action:**
> The sentence in Section 3.2.3 has been rewritten as:
>
> > “Indirect-to-task representation generation methods can access complete multimodal data samples during training, which means they can simulate arbitrary missing modality cases during training by dropping modalities. As a result, they generally achieve better performance than direct-to-task methods, which must handle incomplete multimodal data samples during training.”
>
> ---
>
> ## **Q: Edit the “self-distillationZhang et al. (2019).”**
>
> **A:**
> We thank the reviewer for catching this typo.
>
> **Action:**
> Corrected to:
>
> > “self-distillation (Zhang et al., 2019).”
>
> See Section 4.1.2.

---

> ### Author Response · Authors · 2025-12-10
>
> Dear Reviewer,
>
> I sincerely apologize for the confusion. I had mistakenly left the rebuttal hidden on OpenReview, but it has now been made visible to everyone. Thank you very much for taking the time to review it.
>
> Best regards,
> Authors

---

### Review · Reviewer_p5sV · 2025-11-23

**Summary Of Contributions:**

The authors provided the systematic and comprehensive overview of Deep Multimodal Learning with Missing Modality (MLMM), a subfield focused on building multimodal systems that remain robust when one or more input modalities are absent during training, testing, or deployment.
Motivated by real-world failures--such as sensor malfunctions, privacy-driven data restrictions, environmental interference, hardware inconsistencies, and domain-specific constraints--the paper positions missing modality as a central challenge for dependable multimodal AI across affective computing, medical imaging, robotics, remote sensing, and space exploration.

**S1.** A central strength of the survey lies in its well-structured and fine-grained taxonomy of MLMM approaches.
The authors organize existing methods along two intuitive dimensions--data-level processing and strategy-level design--which naturally lead to four coherent categories: modality imputation, representation-focused techniques, architecture-oriented methods, and model-combination strategies.
Their explanations within each group are generally convincing and thoughtfully reasoned: they clarify how generative models may introduce hallucinated content, why representation-level recovery can mitigate raw-data noise, how attention-based architectures scale effectively but rely on large datasets, why distillation commonly assumes access to complete modality teachers, how graph-based approaches capture rich structural dependencies but struggle with scalability, and how multimodal LLMs can flexibly handle partial inputs despite heavy pretraining requirements.

**S2.** The survey also proposes a second organizational scheme based on the traditional stages of multimodal fusion--early, intermediate, and late--distinguishing approaches that attempt to reconstruct missing modalities from those that operate without recovery.
Their meta-analysis shows that the majority of existing studies emphasize some form of recovery, with intermediate-stage methods being particularly common because they strike a practical balance between representational richness and stability.
In contrast, comparatively fewer works tackle the problem directly through non-recovery strategies.

**S3.** The authors further survey MLMM applications across information retrieval, medical diagnosis, sentiment analysis, multimodal time-series, remote sensing, robotic vision, and autonomous driving, providing an extensive dataset compilation for each domain.
They also analyze the rise of multimodal transformers and multimodal LLMs (e.g., CLIP, BLIP-2, LLaVA, AnyGPT, ImageBind), highlighting how large-scale pretraining implicitly helps models cope with partial modality inputs.

**W1.** However, the survey overlooks an important and rapidly growing domain where missing-modality issues are intrinsic rather than incidental: IoT, pervasive sensing, and distributed real-world sensor networks.
In these systems--ranging from smart homes and industrial monitoring to wearable platforms and environmental sensor grids--modalities fail unpredictably due to battery loss, bandwidth limits, asynchronous sampling, heterogeneous device capabilities, and physical degradation.
These settings represent some of the most persistent and structurally inevitable instances of missing modalities, yet the survey touches only lightly on them and provides limited references or methodological discussion tailored to this context.
As these real-world sensing ecosystems increasingly rely on multimodal fusion (vision + audio + inertial + environmental + network data), MLMM techniques must address challenges such as long-term partial observability, streaming data with asynchronous modalities, multi-agent inconsistencies, and on-device resource constraints--areas that remain underrepresented in the current review.

**Additional Comments:**

NA

**Audience:**

Yes

**Audience Explanation:**

The survey on missing-modality multimodal learning will likely draw significant attention because it offers one of the most comprehensive and systematic overviews of a rapidly expanding research area.
As multimodal models increasingly power applications in vision–language understanding, medical imaging, robotics, human–computer interaction, and pervasive sensing, practitioners frequently encounter scenarios where one or more modalities are absent, degraded, or unavailable.
This survey distinguishes itself by bringing clarity to this methodological landscape through a carefully structured taxonomy spanning modality imputation, representation-centric strategies, architecture-level solutions, and model-combination approaches.
The breadth of coverage--integrating over 300 papers across diverse domains--gives readers a holistic understanding of how missing-modality challenges have been addressed historically and where the field is heading.

Moreover, the paper’s dual categorization framework (methodological perspective and fusion-stage perspective) helps unify previously scattered contributions under a coherent conceptual structure, making it easier for researchers and practitioners to navigate the space, compare methods, and identify gaps.
Its extensive discussion of strengths, limitations, and trade-offs of each method family further enhances its value as a practical reference. Given the accelerating adoption of multimodal large language models and the increasing need for robust, failure-tolerant AI systems, readers across machine learning, computer vision, NLP, medical AI, and IoT sensing communities will find this survey especially timely and informative.

**Claims And Evidence:**

Yes

**Claims Explanation:**

Despite offering a broad survey of multimodal learning under missing-modality conditions, the discussion gives relatively little attention to IoT, pervasive sensing, and real-world distributed sensor ecosystems--domains in which missing modalities are not an occasional anomaly but a fundamental and recurring characteristic of the data environment.
In practical deployments such as smart homes, industrial monitoring platforms, wearable systems, environmental sensor grids, and mobile robotic networks, data streams are inherently heterogeneous, asynchronous, and failure-prone. M
odalities often disappear due to battery depletion, bandwidth limitations, variable sampling rates, intermittent connectivity, hardware degradation, or environmental disruption.

Because these systems routinely operate with partial, degraded, or intermittently available modalities, they present some of the most representative and demanding scenarios for missing-modality learning.
Yet the survey provides only brief coverage of these settings and does not fully articulate the unique challenges they raise--such as cross-device heterogeneity, long-term partial observability, edge-compute constraints, multimodal drift, or the need for continual adaptation in dynamic environments.

As a result, an important class of real-world applications--where MLMM would be not just beneficial but essential--receives limited conceptual framing and methodological treatment.
More explicit discussion of IoT and pervasive multimodal sensing would strengthen the completeness of the field overview, especially given the increasing reliance on multimodal fusion in everyday deployed systems.

**Requested Changes:**

A more thorough discussion of missing-modality learning would be strengthened by incorporating insights from wearable sensing, IoT infrastructures, and pervasive multimodal systems.
Prior work on mobile and wearable sensor networks [1] has long emphasized that sensor noise, fluctuating sampling rates, device failures, and user-dependent variability naturally produce persistent gaps in multimodal streams. Recent analyses of ambulatory wrist-worn devices further illustrate how real-world data are characterized by non-wear intervals, dropout segments, and irregular sensor availability [2].

Studies in multi-sensor human activity recognition similarly highlight the structural difficulties posed by heterogeneous sensing environments. For example, multi-graph modeling approaches [3] show that IMUs, magnetometers, and physiological sensors often sample asynchronously or provide incomplete data, underscoring the importance of techniques that remain effective under partial modality access. Work on robustness and data augmentation for wearable signals [4] likewise demonstrates that signal degradation, corruption, and cross-modality inconsistency can substantially degrade downstream model performance--conditions directly relevant to missing-modality learning.

Complementary research on recovering absent sensor channels provides additional rationale for stronger integration of these domains. Generative reconstruction approaches such as SensorGAN [5] aim to restore missing sensor streams, while recent efforts on lightweight imputation strategies for mobile health systems [6] develop efficient reconstruction pipelines suitable for resource-constrained devices. Collectively, these lines of work make clear that missing modalities are not rare artifacts but a defining characteristic of distributed sensing, ubiquitous computing, and mobile health ecosystems.

Integrating these references would help situate missing-modality learning within the practical settings where it is most essential--demonstrating that MLMM methods must be designed not only for curated multimodal datasets but for real-world sensor networks where modality loss, degradation, and heterogeneity are intrinsic and unavoidable.

**Reference**
- [1] Nweke, Henry Friday, et al. "Deep learning algorithms for human activity recognition using mobile and wearable sensor networks: State of the art and research challenges." Expert Systems with Applications 105 (2018): 233-261.
- [2] Van Der Donckt, Jonas, et al. "Mitigating data quality challenges in ambulatory wrist-worn wearable monitoring through analytical and practical approaches." Scientific Reports 14.1 (2024): 17545.
- [3] Chen, Ling, et al. "A multi-graph convolutional network based wearable human activity recognition method using multi-sensors." Applied Intelligence 53.23 (2023): 28169-28185.
- [4] Jeon, Eun Som, et al. "Role of data augmentation strategies in knowledge distillation for wearable sensor data." IEEE internet of things journal 9.14 (2021): 12848-12860.
- [5] Hussein, Dina, and Ganapati Bhat. "SensorGAN: A novel data recovery approach for wearable human activity recognition." ACM Transactions on Embedded Computing Systems 23.3 (2024): 1-28.
- [6] Hussein, Dina, et al. "Energy-efficient missing data imputation in wearable health applications: A classifier-aware statistical approach." Proceedings of the Thirty-Third International Joint Conference on Artificial Intelligence AI for Good. 2024.

---

> ### Author Response · Authors · 2025-12-05
>
> We sincerely thank the Reviewer for the constructive suggestions. Below, we provide our responses to the weaknesses raised. All corresponding revisions have been incorporated into the updated submission, with changes clearly highlighted for ease of reference.
>
> ---
>
> ## **Q: Insufficient discussion of IoT, pervasive sensing, wearable systems, and distributed multimodal sensor networks as core domains for missing-modality learning.**
>
> **A:**
> We thank the reviewer for identifying this important gap. We agree that IoT ecosystems, pervasive sensing environments, and distributed sensor networks represent some of the most realistic and structurally unavoidable settings for missing-modality problems, and this domain warrants a more thorough and dedicated discussion.
>
> **Action:**
> To address this, we substantially expanded the survey in the following ways:
>
> 1. We inserted the references listed by the Reviewer in the field of pervasive computing for handling missing modalities into the corresponding method categories, and also added several other relevant papers.
> 2. We added a new subsection to highlight the important application area **“Pervasive Computing”** (see **Sec. 6.5**) in the *Applications and Datasets* section.
> 3. **Table 4** was updated to include representative datasets used in corresponding fields.
> 4. We introduced a new subsection discussing the importance of practical missing-modality challenges in real-world systems (see **Sec. 7.8**). In this subsection, we clarified that although notable progress has been made in pervasive computing, current solutions remain far from sufficient—particularly due to the scale, heterogeneity, persistence of missing modalities, and the need for lightweight models in pervasive environments.
>
> ---
>
> All above modifications have been highlighted in the revised submission.

---

### Review · Reviewer_fwck · 2025-11-26

**Summary Of Contributions:**

This paper reviews Deep Learning approaches to Multimodal Systems, more specifically in the context of missing modalities (which is represented by the MLMM acronym).
It reviews 354 papers published between 2012 and 2025, in AI, ML and more content or application focussed titles such as ICASSP, ACM Multimedia or MICCAI.  The review aims to cover various architectural approaches and application areas, which include information retrieval, remote sensing, robotic vision, medical diagnosis and sentiment analysis.
It purports to make three contributions: i) a comprehensive survey of MLMM methods across diverse domains; ii) A novel, fine-grained taxonomy of MLMM methodologies and iii) an in-depth analysis of current MLMM approaches, their challenges, and future research directions.

These contributions are achieved to a variable extent. Contribution ii) is clearly visible and consists in an original and comprehensive taxonomy, even though some details may warrant further discussion such as the inclusion of ‘attention’ as a generic component rather than associated with a specific DL architecture other than a Transformer. Contribution i) is partly achieved, the main question being the scope of the review that might result in an imbalance between most recent approaches and earlier one due to the major evolutions of the discipline (and the growth in submissions across the year which creates another, quantitative, imbalance).  Contribution iii) could be found wanting, primarily for lack of in-depth analysis. In many places the discussion is mostly about categorizing approaches and clustering publications with no real additional conceptual frameworks beyond the above taxonomy, which is not sufficient to that purpose.

The main strengths of the paper consist in the identification of the MLMM topic; the collection of references; the attempted breadth of coverage in terms of application; the generic presentation; the volume of publications reviewed.

Its weaknesses include: a reviewing philosophy which primarily appears as clustering and categorization, with direct comments on individual works and their clustering, and limited analysis frameworks apart from the taxonomy; a lack of clarity on the technical evolution of the discipline; some difficulties in articulating architectures and applications.
An overall weakness of the paper is not to have developed a conceptual framework for multimodality into which to analyze the issue of missing modalities. Here I am making reference to the kind of frameworks that are pervasive in Multimodal research, which discuss relationship between modalities in terms or temporal relations, informational complementarity vs. redundancy, and fusion techniques.

**Additional Comments:**

N/A

**Audience:**

Yes

**Audience Explanation:**

Although the publications in this MLMM topic is more impressive in terms of growth than numbers, which remain limited in the broader ML context, the topic can be of interest to a broader audience, who might encounter MLMM issues as part of their Multimodal ML work or Multimodal application development. The actual extent of this interest depends on improvements and clarifications that would make the findings more substantial and more easily accessible. In the paper’s current form there should be an interest in raw references and their comments, MLMM taxonomy, early categorizations of approaches.

**Broader Impact Concerns:**

This reviewer finds no obvious concerns regarding ethical implications.

**Claims And Evidence:**

No

**Claims Explanation:**

In the context of a review paper, we should consider the claims as corresponding to elements of stated contributions, such as the in-depth nature of the technical analysis or the comprehensive treatment of challenges and future directions.
Because the review spans a long period of technical evolution, from early multimodal fusion networks to contemporary foundation models, MLMM and diffusion-based, the treatment occasionally remains at the level of categorization rather than achieving the in-depth analysis advertised in the contribution section. This is understandable given the breadth of material, yet the result is that the discussion tends to enumerate architectural families without fully articulating their respective motivations, assumptions, or comparative trade-offs.
Related to this, the paper sometimes encounters difficulties in consistently relating architectures to their application domains. Some applications (e.g., captioning, VQA, retrieval, diagnosis, sentiment analysis) introduce intermediate layers or task-specific modules that function as additional interpretive steps between architecture and modality, which makes the classification less clear-cut (this is visible is some medical example and some affective computing examples). The survey describes these applications in detail, but the relationship between architectural choices and the functional requirements of such tasks could be made more explicit.

**Requested Changes:**

Overall, the review would benefit from a conceptual framework for analyzing multimodal architectures and their behaviour under missing-modality conditions. Established perspectives in multimodal research mentioned earlier, such as informational complementarity vs. redundancy, temporal alignment between modalities, or distinctions between early, mid-level, and late fusion could provide the theoretical scaffolding needed to interpret the surveyed literature beyond taxonomic grouping. Such a framework would allow the authors to position individual works not only within categories but also within a set of principled multimodal constructs, thereby offering a more coherent analytical synthesis.

The relationship between applications and architectures needs reworking, whether via the above reframing or through a refined bottom-up approach. Some examples in particular medical ones in 6.2, are barely convincing when they span across so many modalities from HER to omics via many different imaging modalities. Similar comments could be made around what is termed ‘sentiment analysis’ and often seems to refer to various tasks in affective computing: this is such an active field (for instance at ACM Multimedia, which is part of the reviewed venues) that it deserves a more articulate treatment.
In terms of individual section contents and in addition to the above broader comments, sections 6.x necessitate significant rewriting making their MLM tasks more differentiated (think robotic vision and sensing vs. affective computing or medical applications). The discussion section is way too generic and needs to be expanded.

Section 7.3 raises the issue of the balance between recent foundation models and previous approaches: as it is written it would almost make some previous sections appear irrelevant. The authors have quite some room for manoeuvre here as they cannot be blamed for refocusing around specific architectures (which do not have to ba all foundational models).

Section 8, the conclusion, needs substantial expansion. It would also be appropriate to cover some limitations of the paper in view of the challenging nature of the task, setting the limits even to the eventually revised framework for analysis.

---

> ### Author Response · Authors · 2025-12-05
>
> We sincerely thank the Reviewer for the constructive suggestions. Below, we addressed and revised your requests and questions. All changes of text and figures have been highlighted in our new submission for your check.
>
> ---
>
> ## **Q: Missing a conceptual framework for analyzing multimodal architectures**
>
> **A:**
>
> We thank the reviewer for this valuable suggestion. Our taxonomy is motivated by considerations from a **practical multimodal learning pipeline**, focusing on two central directions:
> (1) **how to handle missing modalities from the data perspective**, and
> (2) **how to adapt model architectures to operate robustly under missing-modality conditions**.
>
> We did not attempt to build a taxonomy grounded in theoretical principles, as our primary goal is to provide a **practitioner-oriented** view that reflects how missing-modality issues manifest in real workflows. This aligns with established precedent in multimodal surveys [1,2], which similarly adopt a workflow-oriented perspective.
>
> Overall, while our taxonomy focuses on **data-processing and strategy-design** aspects, we also explicitly relate it back to established multimodal constructs, including **recovery and non-recovery variants of early, intermediate, and late fusion** (see Sec. 5.3).
>
> We further address the reviewer’s point on **informational complementarity and redundancy** in Sec. 7.1, as they play a critical role in guiding accurate modality or representation recovery. These properties are fundamental to multimodal learning rather than artifacts of specific MLMM models.
>
> Regarding **temporal alignment**, we agree it is important; however, alignment typically assumes all modalities are present, which is not the case under missing-modality settings. Since only limited work addresses **streaming or temporally missing modalities**, we discuss this gap in Sec. 7.5 as an essential future direction.
>
> **Action:**
>
> * Added more discussion on informational complementarity and redundancy in Sec. 7.1.
> * Expanded the discussion on temporal alignment in Sec. 7.5.
>
> **References:**
>
> [1] Deep Multimodal Data Fusion, ACM Computing Surveys 2024
>
> [2] Multi-document Summarization via Deep Learning Techniques: A Survey, ACM Computing Surveys 2022
>
> ---
>
> ## **Q: Lack of motivation, assumptions, and trade-offs for various methods**
>
> **A:**
>
> Thanks for the comment. We have introduced the **key assumptions and motivations** at the beginning of each method category and summarized their **advantages and disadvantages** within each section and again in Sec. 5. Our taxonomy is designed from a **practical ML pipeline perspective**, and the motivation of each category reflects how missing-modality issues are handled in real workflows.
>
> ---
>
> ## **Q: Lack of clarity on technical evolution and scope imbalance**
>
> **A:**
>
> We agree. Below are the actions we have taken.
>
> **Action:**
>
> * **Added a stacked bar chart (Figure 18) showing publication trends over the years** according to our taxonomy.
> * **Added Sec. 5.4 “Technical Evolution Discussion”** to summarize major trends over time.
>
> ---
>
> ## **Q: Missing relationship between applications and architectures**
>
> **A:**
>
> Thank you for the comment and sorry for the confusion. Sec. 6.2 is intended only to illustrate **where MLMM problems arise**, why these domains naturally exhibit missing modalities, and which common datasets are used. It is **not** meant to analyze links between architectures and applications.
>
> Based on our survey, **current MLMM methods do not exhibit strong or domain-specific application–architecture relationships**, largely due to the versatility of deep learning.
> For example:
>
> * GAN-based recovery is used in both **breast-cancer prognosis** [1] and **visible–infrared person re-ID** [2].
> * Masked modeling strategies in **action recognition** [3] parallel those in **brain-tumor segmentation** [4].
>
> We have added a discussion explaining **why this link is weak and the implications for MLMM research**.
>
> **Action:**
>
> * **Added a discussion to Sec. 5.4 “Technical Evolution Discussion”.**
>
> **References:**
>
> [1] Generative Incomplete Multi-View Prognosis Predictor for Breast Cancer: GIMPP, IEEE TCBB 2022
>
> [2] Visible-Infrared Person Re-Identification With Modality-Specific Memory Network, IEEE TIP 2022
>
> [3] Towards Good Practices for Missing Modality Robust Action Recognition, AAAI 2024
>
> [4] M3AE: Multimodal Representation Learning for Brain Tumor Segmentation with Missing Modalities, AAAI 2023
>
> ---
>
> ## **Q: Application distinctions corresponding to Sec. 6.x headings**
>
> **A:**
>
> Thank you for the insightful suggestion. We agree that the current Sec. 6.x headings lack clarity in separating domains. Following the reviewer’s recommendation, we reorganized the section so that **affective computing**, **robotic vision and sensing**, **medical applications**, and **other MLMM domains** are clearly distinguished.
>
> **Action:**
>
> * **Updated all Sec. 6.x headings to explicitly reflect task category and application domain.**

---

> ### Author Response · Authors · 2025-12-05
>
> We below continue responding to your question from where we left off in the previous rebuttal.
>
> ---
>
> ## **Q: The discussion section in Sec. 6 needs to be expanded**
>
> **A:**
>
> We thank the reviewer for the suggestion. We agree that Sec. 6 would benefit from a broader discussion of current limitations and open challenges in MLMM research. In particular, we acknowledge the need to elaborate on realistic missingness patterns, dataset availability, and evaluation inconsistencies. If there are any additional works or perspectives you believe warrant further discussion, we would be very happy to consider them and incorporate them into our survey.
>
>
> **Action:**
>
> 1. Discussing the gap between synthetic missing-rate settings and real-world missing-modality patterns observed in pervasive and streaming systems
> 2. Clarifying the lack of standardized protocols for evaluating different missing-modality rates during training and testing.
>
> These updates will provide a more complete and balanced discussion.
>
> ---
>
> ## **Q: Balance between foundation models and earlier approaches**
>
> **A:**
>
> We thank the reviewer for pointing this out. Early MLMM methods were typically developed on small, constrained datasets and domains. In contrast, recent foundation models benefit from large-scale multimodal pretraining and can often rely on learned priors to infer missing information. We have revised **Sec 7.3** to clearly distinguish foundation models from earlier MLMM models.
>
> **Action:**
>
> We rewrote Sec. 7.3 to:
>
> 1. clarify that earlier methods remain relevant and foundational,
>
> 2. highlight the different assumptions and data regimes behind the two families of models, and
>
> 3. ensure the discussion does not overemphasize foundation models at the expense of prior approaches.
>
>
> ---
>
> ## **Q: The conclusion needs to be longer and more comprehensive, and include limitations**
>
> **A:**
>
> Thank you for this helpful suggestion. We agree that the conclusion should more clearly reflect not only the overall contributions of the survey but also the inherent limitations of our taxonomy and analysis.
>
>
> **Action:**
>
> 1. Added more discussion on future directions and limitations of current MLMM techniques (Sec. 8).
> 2. Added limitations of our taxonomy and analysis (Sec. 8).
>
> ---
>
> All above modifications have been highlighted in the revised submission.

---

### Decision · Action_Editor_5d9z · 2026-01-20

**Recommendation:** Accept as is

**Audience:**

Yes

**Audience Explanation:**

Learning with missing modality is growing in importance and researchers in this subfield might benefit from a broader overview that connects various application areas.

**Claims And Evidence:**

Yes

**Claims Explanation:**

The paper presents the first in-depth practitioner-first survey of the subfield of multimodal learning that focuses on learning under the missing modality problems.

The survey was generally appreciated for its utility and the introduced taxonomy.

The submission was significantly improved during the rebuttal phase. In particular, a discussion of IoT devices was added, where the problem is especially pervasive.

Two Reviewers recommended acceptance, while one leaned towards rejecting the paper.

The key negative comment concerned the lack of synthesis. In response, the Authors have added the Technical Evolution subsection and significantly expanded the Open Issues section. It can be argued that the paper doesn’t provide an overall conceptual synthesis of the field. However, I believe it is a useful account of the many changes that the subfield has undergone.

All in all, the work is likely to be of interest to the audience and is a valuable contribution. It is my pleasure to recommend acceptance.

---

> ### Author Response · Authors · 2026-02-03
> **Camera-Ready Version Submitted**
>
> Dear all editors and reviewers,
>
> We have submitted the camera-ready version of the manuscript.
>
> We would like to sincerely **thank** the Action Editor and the Editors-in-Chief for their support throughout the review process. We are also very grateful to the reviewers for their constructive and insightful feedback, which substantially improved the quality and clarity of the paper.
>
> Best,
>
> Authors